# Observed microphysical changes in Arctic mixed-phase clouds when transitioning from sea ice to open ocean

Gillian Young[1], Hazel M. Jones[1], Thomas W. Choularton[1], Jonathan Crosier[2,1], Keith N. Bower[1], Martin W. Gallagher[1], Rhiannon S. Davies[3], Ian A. Renfrew[3], Andy D. Elvidge[4], Eoghan Darbyshire[1], Franco Marenco[4], Phil R. A. Brown[4], Hugo M. A. Ricketts[2,1], Paul J. Connolly[1], Gary Lloyd[2,1], Paul I. Williams[2,1], James D. Allan[2,1], Jonathan W. Taylor[1], Dantong Liu[1], and Michael J. Flynn[1]

[1]Centre for Atmospheric Science, University of Manchester, Manchester, UK.
[2]National Centre for Atmospheric Science, University of Manchester, Manchester, UK.
[3]School of Environmental Sciences, University of East Anglia, Norwich, UK.
[4]Met Office, Exeter, UK.

*Correspondence to:* G. Young (gillian.young@manchester.ac.uk)

**Abstract.** In situ airborne observations of cloud microphysics, aerosol properties, and thermodynamic structure over the transition from sea ice to ocean are presented from the Aerosol-Cloud Coupling and Climate Interactions in the Arctic (ACCACIA) campaign. A case study from 23 March 2013 provides a unique view of the cloud microphysical changes over this transition under cold air outbreak conditions. Both cloud base lifted and cloud depth increased from sea ice to ocean. Mean droplet
number concentrations, $N_{drop}$, also increased from $110 \pm 36 \, cm^{-3}$ over the sea ice to $145 \pm 54 \, cm^{-3}$ over the marginal ice zone (MIZ). Downstream over the ocean, $N_{drop}$ decreased to $63 \pm 30 \, cm^{-3}$. This reduction was attributed to enhanced collision-coalescence of droplets within the deep ocean cloud layer. The liquid-water content increased almost four-fold over the transition and this, in conjunction with the deeper cloud layer, allowed rimed snowflakes to develop which precipitated out of cloud base downstream over the ocean.

The ice properties of the cloud remained approximately constant over the transition. Observed ice crystal concentrations averaged approximately $0.5\text{-}1.5 \, L^{-1}$, suggesting only primary ice nucleation was active; however, there was evidence of crystal fragmentation at cloud base over the ocean. Little variation in aerosol particle number concentrations was observed between the different surface conditions; however, some variability with altitude was observed, with notably greater concentrations measured at higher altitudes (>800 m) over the sea ice. Near-surface boundary layer temperatures increased by $13 \, °C$ from sea
ice to ocean, with corresponding increases in surface heat fluxes and turbulent kinetic energy. These significant thermodynamic changes were concluded to be the primary driver of the microphysical evolution of the cloud. This study represents the first investigation, using in situ airborne observations, of cloud microphysical changes with changing sea ice cover and addresses the question of how the microphysics of Arctic stratiform clouds may change as the region warms and sea ice extent reduces.

# 1 Introduction

Projected increases in mean temperature due to climate change are greater in the Arctic than the mid-latitudes (ACIA, 2005). Arctic surface temperatures are predicted to rise by up to $7°C$ by the end of the $21^{st}$ century (ACIA, 2005). As a consequence of recent warming, observations have shown a prominent decline in sea ice volume over the last thirty years (Serreze et al.,
2007), with record-breaking seasonal melts becoming more frequent (e.g. 2004, 2007 and 2012, Stroeve et al., 2005; Perovich et al., 2008; Parkinson and Comiso, 2013). Observed surface air temperatures have displayed more prominent increases in the winter and spring seasons over the past 100 years (Serreze and Barry, 2011); seasonality which greatly affects the associated sea ice formation and melting processes (e.g. ACIA, 2005; Serreze and Barry, 2011).

It is important to better understand cloud microphysics in the Arctic as clouds contribute significantly towards the Arctic
radiative budget (e.g. Intrieri et al., 2002; Shupe and Intrieri, 2004). Arctic clouds often differ from clouds seen at lower latitudes due to differences in aerosol properties and a unique boundary layer structure (Vihma et al., 2014). Additionally, the sea ice is coupled to the Arctic atmosphere and years of decreased summer sea ice extent have coincided with periods of increased cloudiness and humidity during the spring (Kapsch et al., 2013). The relationship between cloud and sea ice fraction adds complexity to Arctic radiative interactions, as increased cloud cover over a low-albedo ocean would typically act to cool
the atmosphere through increased reflectivity of incident solar radiation (Curry et al., 1996; Shupe and Intrieri, 2004). However, in the Arctic this incoming shortwave (SW) solar radiation is minimal during the autumn through to spring (Curry et al., 1996), allowing the upward longwave (LW) heat fluxes from the surface to dominate (Intrieri et al., 2002; Palm et al., 2010). The small cloud droplets common in Arctic clouds trap upwelling infra-red radiation efficiently, leading to almost twice the amount of total LW than SW radiation at the surface per annum (Curry et al., 1996). Increased springtime cloudiness can therefore lead
to increased trapped LW radiation and surface warming, thus potentially affecting the sea ice melting process.

Single-layer mixed-phase stratocumulus (MPS) clouds are particularly common in the Arctic (e.g. Pinto, 1998; Shupe et al., 2006; Verlinde et al., 2007). Such clouds are sustained by small vertical motions and are characteristically topped with a liquid layer which facilitates ice formation below (Shupe et al., 2006; Vihma et al., 2014). Single-layer MPS are particularly prevalent in the transition seasons (Shupe et al., 2006; Morrison et al., 2012), whereas multi-layered MPS are more common during the
summer (Curry et al., 1988, 1996). It is uncertain how Arctic cloud fractions will evolve with increased global temperatures (Curry et al., 1996), and comprehending their relationship with sea ice extent is key to improving the representation of radiative interactions in numerical models. Palm et al. (2010) used remote sensing techniques to show that cloudiness typically increases over the marginal ice zone (MIZ) and ocean in comparison to over the sea ice, forming deeper cloud layers with greater optical depth over the ocean. This study also identified increased cloud fractions in years with decreased sea ice cover, implying an
important feedback for Arctic warming. Further investigation of cloud properties in the context of surface ice cover could therefore improve both our understanding of the microphysics of high latitude clouds and their dependency on and sensitivity to the surface conditions.

Within global climate models (GCMs), one of the largest sources of uncertainty is our poor understanding of cloud and aerosol processes, and this is particularly an issue in the polar regions (Boucher et al., 2013). The paucity of observations in the

Arctic leads to an inadequate understanding of aerosol-cloud interactions, which in turn impacts our ability to accurately model the cloud microphysics, boundary layer structure, and radiative interactions in this region (Curry et al., 1996). There has been a drive to collect more in situ observations of Arctic MPS over recent decades. Studies such as the Mixed-Phase Arctic Cloud Experiment (M-PACE, Verlinde et al., 2007) and the Indirect and Semi-Direct Aerosol Campaign (ISDAC, McFarquhar et al., 2011) collected in situ aircraft observations over the Beaufort Sea near Barrow, Alaska during the transition seasons; autumn 2004 and spring 2008 respectively. These studies have substantially improved our knowledge of transition season Arctic clouds; however, key questions remain. For example, how does cloud microphysics change with a changing surface? And do Arctic clouds differ with geographical location? Jackson et al. (2012) found a greater mean liquid-water content in clouds over the ocean (during M-PACE) compared with those over the sea ice (during ISDAC), and substantial microphysical differences have been previously identified between cloud observations at three permanent measurement stations in the Canadian Arctic, based on meteorological differences (Shupe, 2011). Given such heterogeneity, studies of other Arctic regions are necessary.

The Aerosol-Cloud Coupling and Climate Interactions in the Arctic (ACCACIA) campaign was carried out to address these questions amongst others. Conducted in the European Arctic in 2013, the ACCACIA project was split into two campaign periods; one in spring (Mar-Apr), the other in summer (Jul-Aug). During the springtime campaign, the Facility for Airborne Atmospheric Measurements (FAAM) BAe-146 atmospheric research aircraft was used to collect high resolution data of cloud and aerosol properties – along with meteorological parameters such as air temperature, humidity, and turbulence – in the Svalbard archipelago off the northern coast of Norway. A primary objective of ACCACIA was to investigate both the microphysical properties of MPS in the European Arctic and their relationship with sea ice cover. In this study, detailed observations from one case study are presented to illustrate the changing microphysical structure of clouds with sea ice extent.

## 2 Instrumentation and data analysis

### 2.1 FAAM aircraft

The FAAM modified BAe 146-301 Atmospheric Research Aircraft (ARA) is fitted with a suite of aerosol, cloud microphysics, and remote sensing instrumentation, detailed by Crosier et al. (2011), Liu et al. (2015), and Lloyd et al. (2015) amongst others. Measurements from these instruments are used here to investigate microphysical properties of clouds in the context of their environment. In this article, all data are expressed as ambient measurements, and number and mass concentrations are not corrected to standard temperature and pressure conditions.

#### 2.1.1 Meteorological instrumentation

The FAAM Core instrument set was active during this campaign (see Renfrew et al., 2008). The GPS-aided Inertial Navigation system and Rosemount temperature sensor are utilised in this study to measure the aircraft's geographical position and the ambient atmospheric temperature respectively. 3D wind components were measured using both a 5-hole turbulence probe and an AIMMS20AQ turbulence probe (Beswick et al., 2008). Dropsondes were released during the campaign to retrieve vertical

profiles of the atmospheric temperature and relative humidity (RH), amongst other properties. Additionally, a downward-facing Leosphere ALS450 lidar provided measurements of cloud top height below the aircraft.

### 2.1.2 Aerosol instrumentation

Sub-micron non-refractory aerosol composition was measured by a Compact Time of Flight Aerosol Mass Spectrometer (C-ToF-AMS, Aerodyne Research Inc., Canagaratna et al., 2007). This instrument has been used extensively in previous aircraft campaigns to characterise such aerosol (e.g. Morgan et al., 2010). Black carbon loadings were monitored with a Single Particle Soot Photometer (SP2, Droplet Measurement Technologies, DMT) and its usage during the ACCACIA campaign is discussed by Liu et al. (2015).

Fine-mode aerosol particle concentrations (spanning particle diameters, $D_P$, 3 nm - 3 μm) were measured using a TSI 3786-LP ultrafine condensation particle counter (CPC). A Passive-Cavity Aerosol Spectrometer Probe (PCASP 100-X, DMT, Rosenberg et al., 2012) was used to count and size accumulation-mode aerosol particles of sizes 0.1 μm to 3 μm. Particle samples (of sizes ∼0.1 μm – 10 μm) were collected on Nuclepore polycarbonate filters exposed from the aircraft for compositional analysis (Young et al., 2016). Additionally, number concentrations and size distributions of aerosol particles and cloud droplets (of sizes 0.6 μm to 50 μm) were measured using the Cloud Aerosol Spectrometer with Depolarisation (CAS-DPOL, DMT, Baumgardner et al., 2001; Glen and Brooks, 2013).

### 2.1.3 Cloud microphysical instrumentation

Size-resolved cloud droplet concentrations (3 μm < $D_P$ < 50 μm) were measured with the Cloud Droplet Probe (CDP-100 Version 2, DMT, Lance et al., 2010). These measurements are used to derive the liquid-water content (LWC) of the observed cloud in this study, and this measure is used to distinguish between in- and out-of-cloud observations (using a threshold of $\leq 0.01\,\mathrm{g\,m^{-3}}$ for the latter). Bulk liquid water measurements were also made using a hot-wire (Johnson-Williams) probe: these data compared well, yet there were signal lag issues when exiting cloud with the hot-wire probe. Therefore, the CDP measurement is solely used for the analysis detailed herein. Additionally, these CDP data are used to compute the mean cloud droplet effective radius within the cloud layers.

The 2-Dimensional Stereo particle imaging probe (2DS, SPEC Inc., Lawson et al., 2006) and Cloud Imaging Probes (CIP15, Baumgardner et al., 2001, and CIP100, DMT) are wing-mounted optical array shadow probes (OAPs) used here to investigate the ice phase of the clouds observed. The 2DS images with 10 μm resolution over a size range of 10 μm to 1280 μm, whilst the CIP 15 and CIP 100 provide 15 μm and 100 μm resolution from 15 μm to 930 μm and 100 μm to 6200 μm respectively. The CIP15 also provides additional information with 3-level grey-scale image intensity data, used to improve the correction of over sizing due to depth of field errors.

Processing and analysis of these OAP data has been discussed previously (Crosier et al., 2011, 2014; Taylor et al., 2016). Here, we follow the same data processing methodology as Taylor et al. (2016). Particle phase was established by segregating imaged particles into categories based on their circularity (Crosier et al., 2011); highly irregular particles were classified as ice crystals, whilst circular images were classified as cloud or drizzle drops, dependent on size. Image reconstruction was not used

to extend the size ranges of the optical array probes. Phase identification of small particles (<80 μm) could not be conducted due to the low resolution of the 2DS and CIP15 in this limit. Small particles measured by the OAPs are not considered in detail due to this phase uncertainty; therefore, CDP measurements are solely used to investigate small cloud particles. These are assumed to be liquid cloud droplets, and the potential contribution of small ice particles (<80 μm) is not examined.

Finally, 8-bit images of cloud particles were taken with 2.3 μm spatial resolution using the Cloud Particle Imager (CPI, SPEC Inc. Lawson et al., 2001). However, the CPI is not used quantitatively in this study: the small sample volume introduces error into the measurements, manifesting as high local particle concentrations in regions of low ambient number concentrations (Lawson et al., 2001).

## 2.2    Additional data

Derived cloud top temperature from MODIS satellite retrievals and AVHRR visible satellite imagery are used to illustrate cloud spatial structure and distribution. Additionally, sea ice fraction from NASA's National Snow and Ice Data Centre (NSIDC), derived from passive microwave brightness temperatures (Peng et al., 2013), and the approximate ice fraction from the Met Office Unified Model (MetUM) are used to contextualise the in situ observations.

## 3    B762: Case study

Flight B762 took place on 23 March 2013. It was a two part flight starting and ending in Kiruna, Sweden, and re-fuelling at Longyearbyen, Svalbard, Norway. Section 1 of the flight was a continuous high-altitude straight, level run (SLR) at approximately 8000 m, where the lidar was used to sample the cloud structure below. A number of dropsondes were released during this section and the release locations (shown in Fig. 1) allowed for measurements of the atmospheric structure over the varying surface conditions, i.e. open ocean, MIZ, and sea ice. The MIZ occurred between approximately 75-76.5°N, north of which a
continuous sea ice pack was present. In this study, the MIZ is approximated by NSIDC ice fractions between 10% and 90%, as indicated in Fig. 1. Other springtime ACCACIA flights were also designed to investigate changes in atmospheric properties over the transition between sea ice and the ocean; however, flight B762 was the only case which made detailed observations of cloud microphysics over both the sea ice and ocean, and over the transition in between.

Section 2 was split into three parts: a series of SLRs at various altitudes over the sea ice; a sawtooth profile transitioning from
sea ice to ocean; and a second set of SLRs over the sea. The flight was designed to investigate the variation in boundary layer structure and cloud over sea ice and the ocean. Low visibility prevented the second set of runs being completed as planned; however, good data coverage of the cloud over the ocean was still achieved.

## 3.1    Local atmospheric conditions

Cloud layers were observed with the lidar during section 1 of flight B762 (Fig. 2). A continuous layer was observed, where
cloud top descended from approximately 1900 m to 1100 m with increasing latitude. Evidence of a second, lower altitude cloud layer can be seen at high latitudes (500 m at 76.5 °N) through breaks in the continuous layer. The 500 m cloud was not observed

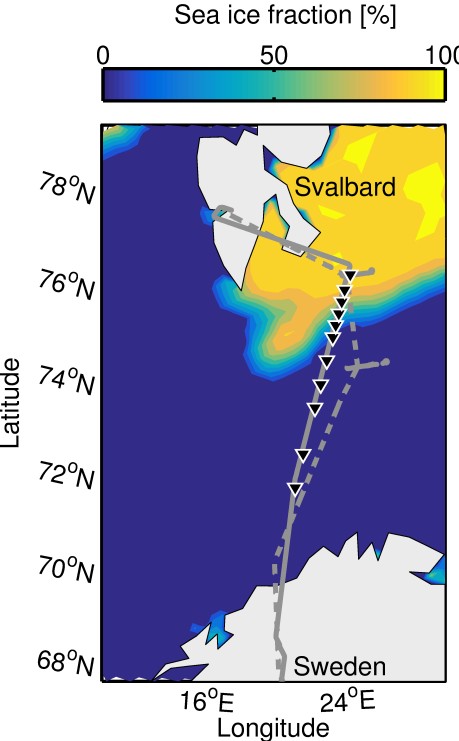

**Figure 1.** Flight track for B762 on 23 March 2013 over sea ice fraction (shading). Section 1 of the flight (grey, solid) was conducted at a high altitude, where 11 dropsondes (black triangles) were released. Section 2 of the flight (grey, dashed) conducted straight, level runs over the sea ice and open water, with a sawtooth profile over the transition region.

at lower latitudes along the flight path sampled (between approximately 73 °N and 73.5 °N), as indicated by the surface echo measured during breaks in the cloud above. An intermittent, high altitude cirrus layer (with optical depth ∼0.5) was seen at various levels from 3000 – 8000 m at higher latitudes on the approach to Longyearbyen. However, these data cannot be used to indicate the spatial extent of these structures as measurements were only made along the flight path.

5    The structure of the lower troposphere was sampled extensively in section 1 by the 11 dropsondes marked in Fig. 1. A summary of key information from each dropsonde is listed in Table 1. These measurements were collected approximately two hours before and to the west of the in situ cloud observations of section 2; however, the dropsonde and lidar measurements provide a good indication of the structure of the atmosphere during this study.

Figure 3 indicates that the boundary layer structure varied with latitude. Figures 3**B** and **D** show vertical profiles of Θ and RH

10   data obtained from dropsondes #5 and #11, collected at latitudes comparable to the in situ aircraft runs of section 2. A double potential temperature inversion can be seen over the sea ice (#11), whereas the temperature profile over the ocean indicates that the boundary layer was well-mixed and coupled to the surface (#5). The double temperature inversion over the sea ice is

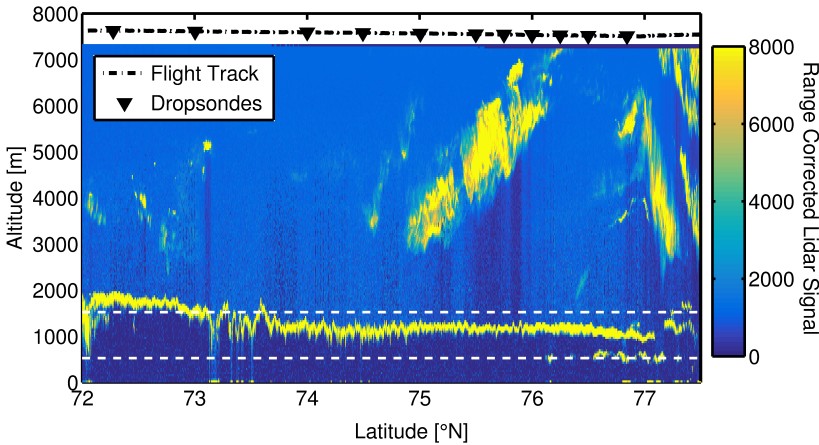

**Figure 2.** Lidar signal from section 1 of the flight. Aircraft altitude is indicated (black) and each dropsonde release point is marked (downward-facing triangles). White dotted lines are shown at 500 m and 1500 m to ease comparison with the in situ observations.

**Table 1.** Key dropsonde information.

| Sonde | Latitude [°N] | Longitude [°E] | Temperature[a] [°C] | Surface Condition[b] |
|:---:|:---:|:---:|:---:|:---:|
| 1 | 72.2 | 21.6 | -3.4 | Ocean |
| 2 | 72.9 | 22.2 | -4.4 | Ocean |
| 3 | 73.9 | 23.3 | -7.3 | Ocean |
| 4 | 74.4 | 23.9 | -9.3 | Ocean |
| 5 | 75.0 | 24.5 | -10.5 | Ocean |
| 6 | 75.4 | 25.2 | -12.3 | Ocean |
| 7 | 75.7 | 25.5 | -13.3 | MIZ |
| 8 | 75.9 | 25.9 | -14.2 | MIZ |
| 9 | 76.2 | 26.2 | -14.7 | Sea ice |
| 10 | 76.4 | 26.6 | -15.0 | Sea ice |
| 11 | 76.8 | 27.3 | -16.4 | Sea ice |

MIZ: Marginal Ice Zone.

[a] Near-surface ambient atmospheric temperature.

[b] Based on NSIDC daily average sea ice fraction.

mirrored by twin RH peaks measured at the corresponding altitudes (∼84%, 500 m and 1100 m, #11, Fig. 3**D**), suggesting the presence of two cloud layers <1500 m below a dry region above the boundary layer (∼1500-2000 m). At lower latitudes over the ocean (74 °N), a single, deep layer (88%) was observed between approximately 300 m and 1200 m. A single temperature inversion was measured by the dropsondes at approximately 1300 m at this latitude. These cloud layers measured by the

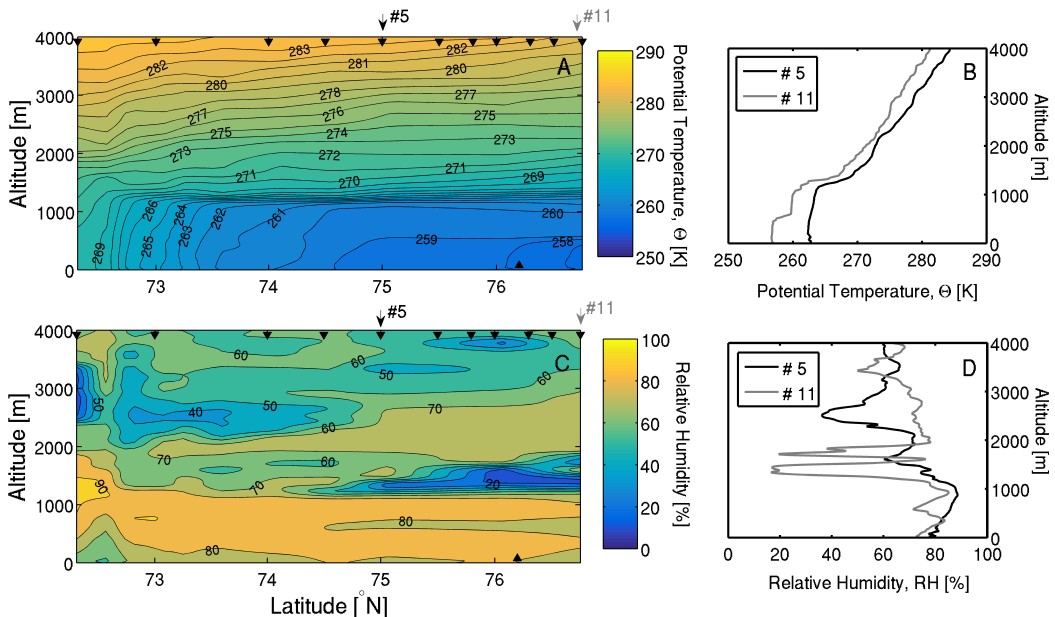

**Figure 3.** Contour figures of potential temperature ($\Theta$, **panel A**) and relative humidity (RH, **panel C**) using data from the 11 dropsondes released on approach to Longyearbyen, Svalbard. Dropsonde release locations (downward-facing triangles) and the approximate sea ice edge (upward-facing triangle) are marked. Profiles of $\Theta$ (**panel B**) and RH (**panel D**) from dropsondes #5 and #11 are also shown due to their comparable latitudes to the in situ observations (see Table 1). The positions of these dropsondes relative to the others are indicated above **panels A** and **C**.

dropsondes are in accordance with the lidar data (Fig. 2). However, these data are affected by a dry bias of approximately 15-20%, where in-cloud RHs of 84% in dropsonde #11 and 88% in dropsonde #5 were measured. Dry biases have been observed in dropsonde data in previous studies (e.g. Ralph et al., 2005), and have been attributed to a slow response time at low temperatures (Poberaj et al., 2002; Miloshevich et al., 2004). Sondes are particularly prone to these response issues when
5   descending from a dry region into a cloudy region (Wang, 2005), which was the case for both of our considered dropsondes.

    Near-surface temperatures sampled by the dropsondes were approximately 13 °C colder to the north over the sea ice (-16.4 °C at 76.8 °N) than over the ocean to the south (-3.4 °C at 72.2 °N, see Table 1). Between the latitudes of the in situ measurements (dropsondes #5 and #11), the difference in near surface temperature is approximately 6 °C.

    Satellite imagery was examined to provide lateral context for the dropsonde and lidar measurements. Figure 4 displays
10   AVHRR visible satellite imagery in panel A and the derived cloud top temperature from MODIS satellite data in panel B. The flight track of section 2 is overlaid to indicate the regions sampled with the aircraft. The high altitude cirrus layer indicated by Fig. 2 can be seen with both of these data. This cirrus cloud was to the north west of the main science region investigated,

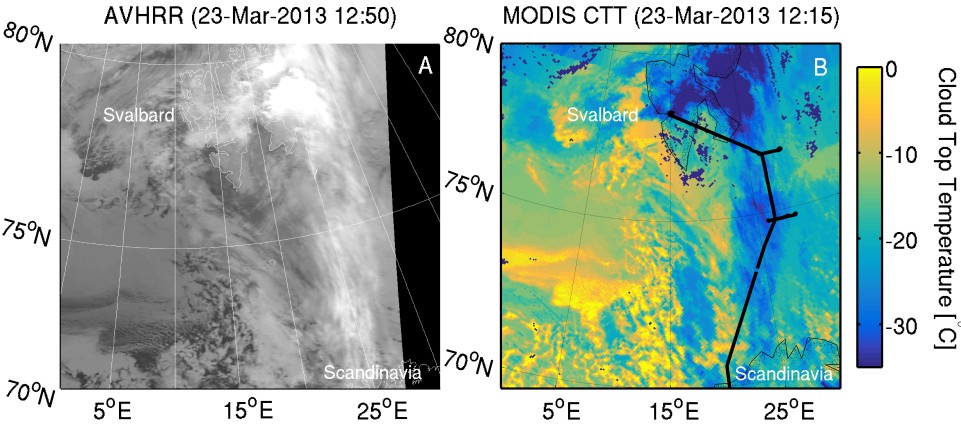

**Figure 4.** AVHRR visible satellite image (**panel A**) and cloud top temperature (CTT) derived from MODIS retrievals (**panel B**) at times close to the start of section 2 of B762. Section 2 of the flight track is indicated (black) in panel B.

closer to Spitsbergen. At the locations sampled during the aircraft runs (Fig. 4**B**), there is no clear indication of a higher cirrus cloud layer and the cloud top temperature is approximately -18 °C and -23 °C over the sea ice and ocean regions respectively.

A weak low pressure system was present to the east during the sampling period; however, conditions were dominated by high pressure to the west, causing a northerly flow of air off the sea-ice. Cold-air outbreak conditions – with wind speeds of $\sim 10\,\mathrm{ms^{-1}}$ measured within the boundary layer – were maintained for the duration of the science period. Aircraft measurements were made through a band of cloud orientated in the N-S direction, influenced by this northerly flow. Back trajectory analyses (shown in the Supplement, Fig. S1) also show that the sampled air came from the north having travelled from northern Canada and/or Greenland, depending on the period of interest.

## 4   In situ observations

### 4.1   Cloud microphysics

#### 4.1.1   Sea ice

Section 2 of the flight began in Longyearbyen, Svalbard and ended in Kiruna, Sweden. A series of SLRs were performed on an easterly or westerly heading, at an approximately constant latitude. Details of each run are listed in Table 2. Run 7 finished early due to instrument icing as a result of flying in the supercooled mixed phase cloud layer. No additional runs after run 8 were possible as visibility was severely reduced due to below cloud haze. A time series of the microphysical observations is shown in Fig. 5.

Figure 6 shows the droplet and ice crystal number concentrations ($N_{drop}$, $N_{ice}$) measured over the sea ice by the CDP and 2DS. These measurements indicate the presence of a mixed-phase cloud between approximately 300 m and 700 m, with

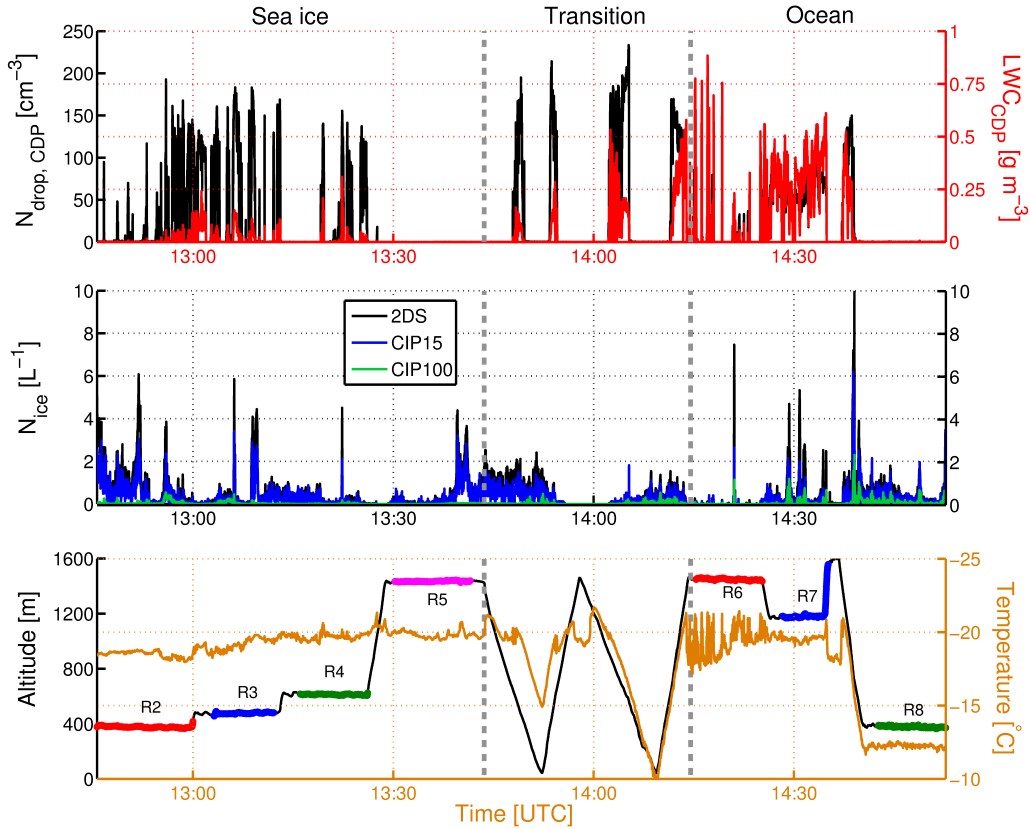

**Figure 5.** Time series of data collected during the science period of ACCACIA flight B762. **Top row:** CDP droplet number concentration (black) and derived liquid-water content (red). **Second row:** 2DS (black), CIP15 (blue), and CIP100 (green) ice number concentration. **Bottom row:** GPS altitude (black), with individual SLRs noted in colour, and temperature measured by the Rosemount de-iced temperature sensor (orange). SLR colours relate to data shown in Figs. 6 and 9. Sea ice, transition, and ocean regions are indicated above the top row.

mean droplet and ice number concentrations of approximately $90\,\mathrm{cm}^{-3}$ and $1\,\mathrm{L}^{-1}$ respectively at $\sim400\,\mathrm{m}$. Mean droplet number concentrations varied with altitude in cloud, with overall average (and standard deviation) of $110\pm36\,\mathrm{cm}^{-3}$. 2DS ice concentrations agree well with the CIP15 (shown in Fig. 5). Derived mean CDP LWC peaks at $\sim0.05\,\mathrm{g\,m}^{-3}$ at $400\,\mathrm{m}$, where the mean temperature is -19°C. For the majority of the sea ice cloud layer sampled, rimed crystals dominated. Other habits
5   such as columns and dendrites were also prevalent. The double temperature inversion suggested by the sea ice dropsondes in Fig. 3 can also be seen here at $600\,\mathrm{m}$ and $1100\,\mathrm{m}$, though not as clearly. The secondary cloud layer at $\sim1000\,\mathrm{m}$, indicated by the dual RH peaks in Figs. 3**C** and **D**, is observed and is dominated by ice; however, it is not as extensive as the main layer. This layer likely dissipated somewhat from the time of the dropsonde measurements, removing the in-cloud liquid indicated by the lidar measurements (Fig. 2) through the Wegener-Bergeron-Findeisen (WBF) mechanism. For the purpose of this study,
10   this sea ice cloud ($300\,\mathrm{m}$–$700\,\mathrm{m}$) is treated as a single-layer MPS.

**Table 2.** Straight and level run information. Values quoted are arithmetic mean quantities, with $1\sigma$ quoted in brackets.

| Run | Start time [UTC] | End time [UTC] | Direction | Altitude[a] [m] | Temperature[b] [°C] | % in cloud[c] | Latitude [°N] | Surface Condition |
|-----|------------------|----------------|-----------|-----------------|---------------------|---------------|---------------|-------------------|
| 2 | 12:45:39 | 13:00:00 | W to E | 377 (6) | -18.5 (0.2) | 11.7 | 76.8 | Sea ice |
| 3 | 13:03:07 | 13:12:06 | E to W | 477 (5) | -19.2 (0.4) | 27.0 | 76.8 | Sea ice |
| 4 | 13:16:03 | 13:26:10 | W to E | 612 (3) | -19.7 (0.2) | 17.8 | 76.8 | Sea ice |
| 5 | 13:30:15 | 13:41:33 | E to W | 1435 (4) | -19.9 (0.3) | 0 | 76.8 | Sea ice |
| 6 | 14:15:20 | 14:25:16 | W to E | 1449 (6) | -19.4 (0.9) | 14.7 | 74.8 | Ocean |
| 7 | 14:28:19 | 14:35:09 | E to W | 1190 (56) | -19.6 (0.3) | 92.0 | 74.8 | Ocean |
| 8 | 14:42:39 | 14:52:46 | W to E | 378 (5) | -12.3 (0.1) | 0 | 74.8 | Ocean |

[a] Derived from GPS measurements.

[b] Ambient temperature measured with the Rosemount de-iced temperature sensor.

[c] In cloud defined as when CDP LWC $\geq 0.01\,\mathrm{g\,m^{-3}}$.

During these sea ice SLRs, the aim was to measure below, in and above the cloud layer. The lowest altitude case (run 2) was carried out in a haze layer present between cloud base and the surface. This haze was only measured over the sea ice region. As described by Young et al. (2016), aircraft filters were exposed during this run, and a silicate dust concentration of $\sim 0.4\,\mathrm{cm^{-3}}$ was measured. No vertical profiles of mineral dust were obtained, and these data are only valid below-cloud, at approximately 380 m, over the sea ice. The size distribution of this mineral dust, in conjunction with the cloud top temperature (-19.7°C), was used to evaluate the Niemand et al. (2012, hereafter N12) primary ice nucleation parameterisation. Due to turbulent motions within the cloud, ambient temperatures could not be used as an indicator of INP/ice crystals at specific altitudes; therefore, the coldest in-cloud temperature was used to provide an upper limit of predicted ice concentrations within the cloud. Predicted ice number concentrations were approximately $0.7\,\mathrm{L^{-1}}$. Dust concentrations of double and triple that measured were also used to evaluate N12 to test sensitivity to this input. The shape of the dust surface area distribution was maintained and the number concentration in each bin was scaled accordingly. Dust number concentrations $>0.5\,\mu\mathrm{m}$ were also used to evaluate the DeMott et al. (2015, hereafter D15) parameterisation, predicting $0.02\,\mathrm{L^{-1}}$. Additionally, number concentrations of all measured aerosol particles $>0.5\,\mu\mathrm{m}$ from the aircraft filters, PCASP, and CAS-DPOL were used to evaluate the DeMott et al. (2010, hereafter D10) and Tobo et al. (2013, hereafter T13) parameterisations. Predicted ice number concentrations were $1.90\,\mathrm{L^{-1}}$ and $1.10\,\mathrm{L^{-1}}$, using PCASP data, respectively. Inputs and outputs of these four parameterisations are detailed in Table 3.

Longitudinally-separated data from runs 2, 3 and 4 are displayed in Fig. 7. The cloud was observed to be spatially inhomogeneous. Cloud particle concentrations increase at similar geographical locations indicating that the same cloud layer was sampled at the different altitudes. Run 5 was conducted above the cloud layer to characterise aerosol size distributions and composition. However, ice crystals were observed, and images collected by the CPI towards the end of this run are shown in Fig. 8. Pristine bullet rosettes were observed, indicating that these crystals had fallen from a greater height without interaction with liquid cloud. Bullet rosettes were observed at the western fringes of the sea ice cloud with the 2DS and CIP15 ($\sim$27,°E,

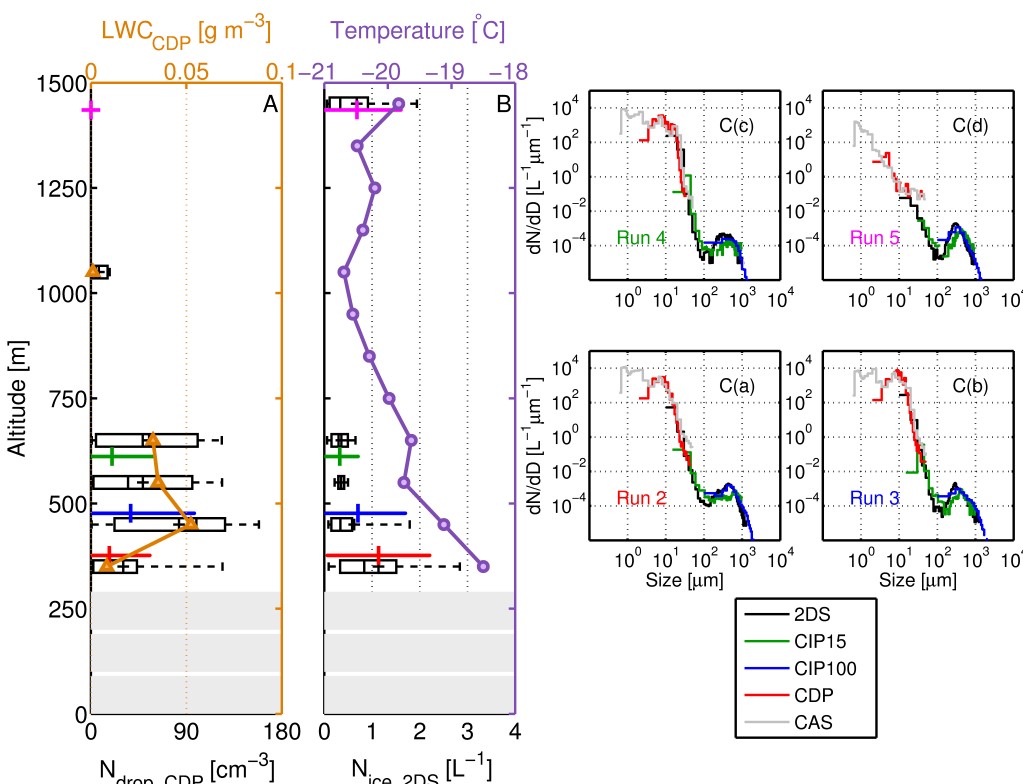

**Figure 6.** Microphysics summary of averaged observations over the sea ice. **A:** CDP droplet number concentration (boxes, black) with derived liquid-water content overlaid (orange). **B:** 2DS ice crystal number concentration (boxes, black) with mean temperature measured overlaid (purple). Only CDP and 2DS data $>0.5 \, \mathrm{cm}^{-3}$ and $>0.05 \, \mathrm{L}^{-1}$ respectively have been included. Box edges represent the $25^{\mathrm{th}}$ and $75^{\mathrm{th}}$ percentiles, and the median and mean values are denoted by | and + respectively. Altitudes not sampled are blocked out ($\leq 300$m). **A** and **B**: number concentrations from each SLR are shown in colour at each corresponding altitude (run 2: red, run 3: blue, run 4: green, run 5: magenta). Arithmetic means are indicated (|) with each horizontal bar extending to $\pm 1\sigma$. **C(a-d):** Number size distributions (dN/dD) from each SLR over the sea ice (runs 2-5 respectively). Legend refers to panel **C** only.

Fig. 7); however, some crystal aggregates were also observed with the CPI in the main cloud, though these were very few in number. Due to the dominance of large rimed ice crystals, it is difficult to conclusively state if appreciable concentrations of these bullet rosettes precipitated into and interacted with the main cloud layer considered.

Figures 6**C(a-d)** show the number size distributions measured along each SLR carried out over the sea ice. A droplet mode at $\sim 10 \, \mu$m can be seen in Figs. 6**C(a-c)** – corresponding to the in-cloud runs (runs 2-4) – with the CAS-DPOL and CDP measurements. This mode is distinctly missing from the run 5 data; negligible droplet concentrations were observed at this altitude, with ice crystals dominating overall particle concentrations.

**Table 3.** Summary of inputs to and evaluations of the N12 (Niemand et al., 2012), D15 (DeMott et al., 2015), D10 (DeMott et al., 2010), and T13 (Tobo et al., 2013) parameterisations. Silicate dust concentrations – derived from filter analysis presented by Young et al. (2016) – are used to evaluate N12 and D15. No filter data are available over the ocean.

| Surface | Temperature [°C] | Aerosol Input | $N_{aerosol}$ [cm$^{-3}$] | $N_{ice}$ [L$^{-1}$] | | | |
|---|---|---|---|---|---|---|---|
| | | | | N12 | D15 | D10 | T13 |
| Sea ice | -19.7 | Filter[a] | 0.4 | 0.7[b] | - | - | - |
| | | Filter[a,c] | 0.2 | - | 0.02 | - | - |
| | | 2×Filter[a] | 0.7 | 1.4[b] | - | - | - |
| | | 3×Filter[a] | 1.1 | 2.1[b] | - | - | - |
| | | Filter[c] | 0.6 | - | - | 0.9 | 0.07 |
| | | PCASP[c] | 2.00 | - | - | 1.90 | 1.10 |
| | | CAS-DPOL[c] | 6.85 | - | - | 3.31 | 19.7 |
| Ocean | -20.1 | PCASP[c] | 2.72 | - | - | 2.23 | 2.66 |
| | | CAS-DPOL[c] | 1.11 | - | - | 1.28 | 0.34 |

[a] Silicate dust concentration.

[b] Derived frozen fraction applied to dust distribution.

[c] Particle concentration $>0.5\,\mu$m.

### 4.1.2 Ocean

Figure 9 shows the droplet and ice crystal number concentrations for the ocean section of the flight. Over the ocean, the cloud layer extends from 700 m to 1500 m. CDP LWC displays a more consistent profile in this section, with a mean value of ∼0.3 g m$^{-3}$ measured between approximately 1100 m and 1400 m. Mean droplet number concentrations were again variable with altitude, with an overall average (and standard deviation) of 63 ± 30 cm$^{-3}$. 2DS ice crystal data do not follow the same trend as the droplet data, with variable concentrations measured at each altitude bin.

Run 7 was conducted within the cloud layer and probe icing was noted. The ice number concentrations from this run are not substantially greater than any of the others, suggesting this icing problem may not have greatly affected the measurements; however, there is an increased CIP100 mode within the number size distributions (at sizes >100 µm, Fig. 9**B**) which is not mirrored by the CIP15.

This cloud is more homogeneous in the liquid phase than the layer measured over the sea ice (Fig. 10), with consistent droplet concentrations and LWC values (∼70 cm$^{-3}$ and 0.3 g m$^{-3}$ respectively) measured with changing longitude during each run. As with the sea ice SLRs, a clear droplet mode is visible at approximately 10 µm in runs 6 and 7 (Figs. 9**C(a,b)**). This mode is not clear in run 8; this run was carried out at low altitude below cloud to collect aerosol data. However, as with run 5, some ice was measured by the 2DS, CIP100, and CIP15 (Figs. 7, 9, 10). Images from the CIP100 during run 8 are shown in Fig. 11; large dendritic crystals are present, with notable riming, of sizes ∼1-1.6 mm. Their size and structure suggests interaction with cloud droplets within cloud, subsequent growth, and eventual precipitation as snow.

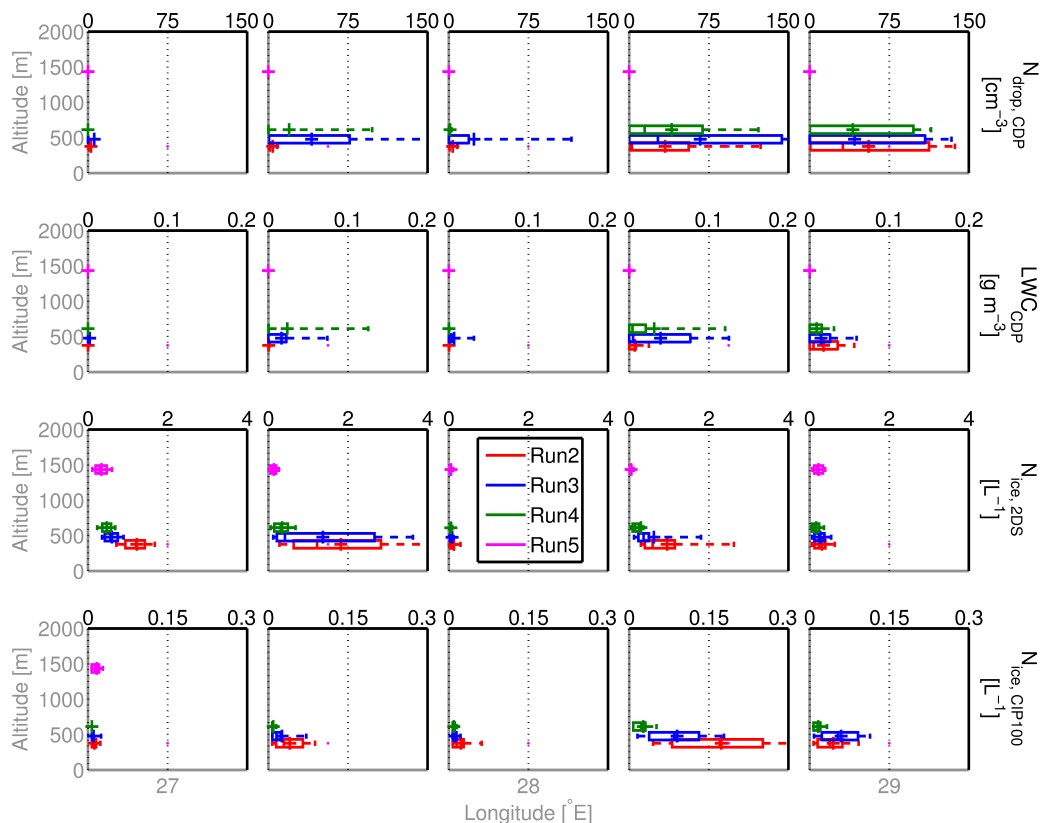

**Figure 7.** Percentile plots of CDP droplet number concentration (**top row**), CDP liquid-water content (**second row**), 2DS ice number concentration (**third row**) and CIP100 ice number concentration (**bottom row**) measured over the sea ice. Each column represents a different longitude bin, moving from west (left) to east (right). Data are plotted against altitude (grey axis), and are coloured differently dependent on the SLR at which the data was measured, as indicated in the legend. Data correspond to the scale on the top x-axis of each segment.

High ice number concentrations were measured between 700 m and 900 m at cloud base. Size distributions from the microphysics probes during this period are shown in Fig. 9**C(d)**: an enhanced secondary mode of ice crystals $\geq 100\,\mu m$ is observed, along with a broadened CDP distribution. The mean temperature measured was approximately -16 °C and numerous large dendrites were observed; large, fragile crystals which may fragment easily upon collision. CIP100 images of these crystals are shown in Fig. S2. 2DS ice crystal concentrations increase to a mean value of approximately $5\,L^{-1}$; much greater than the mean concentration observed within the mixed cloud layer ($\sim 1\,L^{-1}$).

Below-cloud aerosol measurements were again used to evaluate the D10 and T13 parameterisations in conjunction with the cloud top temperature (-20.1 °C). Using PCASP data as input, predicted INP concentrations were $2.23\,L^{-1}$ and $2.66\,L^{-1}$ respectively (see Table 3). No filter data are available over the ocean; therefore, N12 and D15 could not be evaluated.

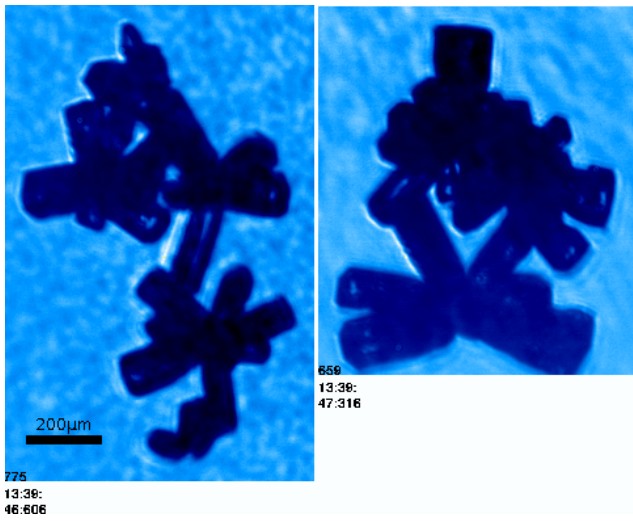

**Figure 8.** CPI ice crystal images from run 5. Time stamps are indicated below each image.

**Table 4.** Sawtooth profile information.

| Profile | Time [UTC] | | Altitude [m] | | Latitude [°N] | | Temperature [°C] | |
|---|---|---|---|---|---|---|---|---|
| | Start | End | Start | End | Start | End | Min | Max |
| 5 | 13:43:37 | 13:52:15 | 1423 | 47 | 76.6 | 76.1 | -21.2 | -15.0 |
| 6 | 13:52:15 | 13:57:50 | 43 | 1450 | 76.1 | 75.8 | -20.7 | -14.9 |
| 7 | 13:57:50 | 14:09:27 | 1459 | 42 | 75.8 | 75.1 | -21.7 | -9.4 |
| 8 | 14:09:28 | 14:14:32 | 43 | 1469 | 75.1 | 74.8 | -21.3 | -9.3 |

### 4.1.3 Transition region

Several profiles were flown in a sawtooth over the transition region from the sea ice to open ocean. Profile 5 was conducted over sea ice; profiles 6 and 7 were over the MIZ; and profile 8 was over the ocean (see Table 4).

Figure 12 shows the CDP droplet number concentrations and derived LWC from each profile, with altitude and mean droplet
5  effective radius, $R_{eff}$, overlaid in the top and bottom rows respectively. These data show a clear lifting and deepening of the cloud layer when transiting from sea ice to open ocean. Mean in-cloud droplet number concentrations increase through the transition, peaking at $145 \pm 54 \, \mathrm{cm}^{-3}$ during profile 7. $N_{drop}$ then begins to decrease in profile 8 ($120 \pm 33 \, \mathrm{cm}^{-3}$). The corresponding mean LWC and droplet effective radii increase from $0.1 \, \mathrm{g \, m}^{-3}$ to $0.4 \, \mathrm{g \, m}^{-3}$ and $5 \, \mathrm{\mu m}$ to $8$-$10 \, \mathrm{\mu m}$ respectively over the transition from sea ice to ocean.

10  Figure 13 shows the 2DS and CIP100 ice concentrations measured over the transition region. Ice number concentrations measured by each instrument remain consistent over the transition, with a mean number concentration of approximately $0.1 \, \mathrm{L}^{-1}$

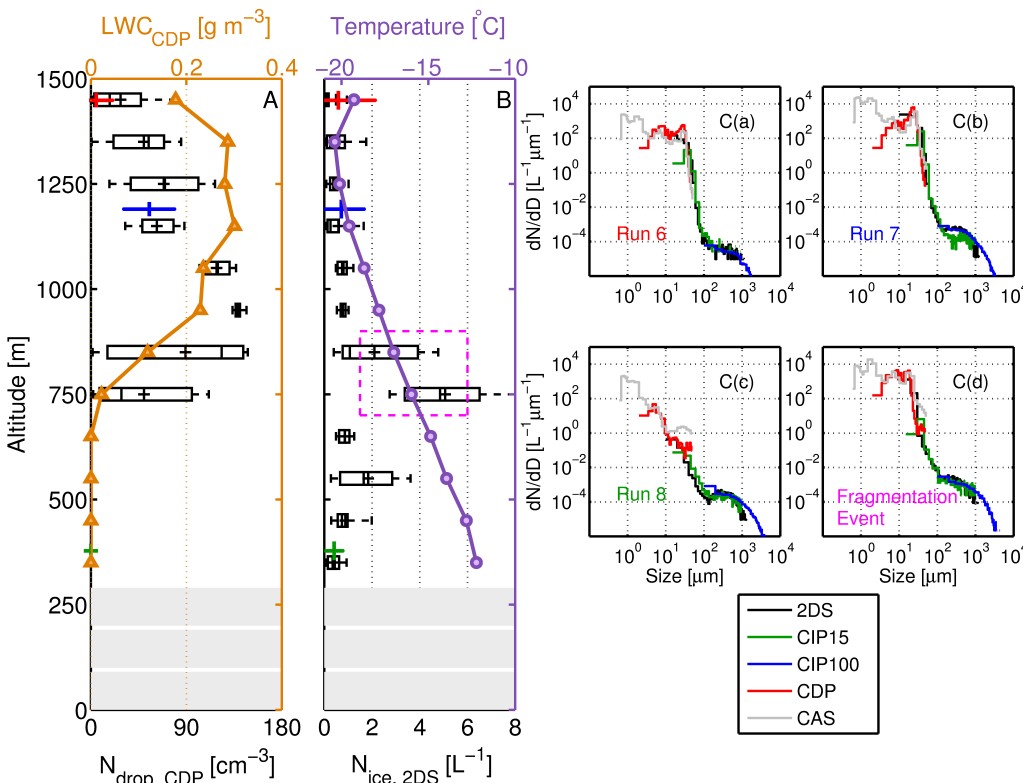

**Figure 9.** Microphysics summary of averaged observations over the ocean. Data are displayed similarly to Fig. 6. In panels **A** and **B**, data from each SLR are again shown in colour at each corresponding altitude (run 6: red, run 7: blue, run 8: green) as before. **C(a-d):** Number size distributions (dN/dD) from each SLR over the ocean (runs 6-8). Legend refers to panel **C** only. High ice number concentrations at cloud base are labelled as a fragmentation event (see Sects. 4.1.2 and 5.3).

measured by the CIP100 over the sea ice and ocean. A slight decrease in the mean 2DS ice concentration can be seen from sea ice to ocean, from approximately $1.5\,\mathrm{L}^{-1}$ to $0.5\,\mathrm{L}^{-1}$. The lifting and deepening of the liquid cloud layer seen in Fig. 12 is not apparent with these ice data. The CIP100 in particular shows a contrasting trend; increasing concentrations below cloud toward the surface suggests precipitation as snow from the cloud layers above. The measured concentrations marginally increase over 5 the ocean, and this precipitation is observed over a greater altitude range due to the lifting of cloud base.

The double temperature inversion indicated in the dropsonde data (Fig. 3) can be viewed in the first two profiles. The lower inversion is eroded to produce a clear, single inversion at $\sim$1400 m during the last profile over the ocean. The gradient of the temperature profile decreases over the ocean due to surface warming, whilst the cloud top temperature remains approximately -20 °C with the changing surface.

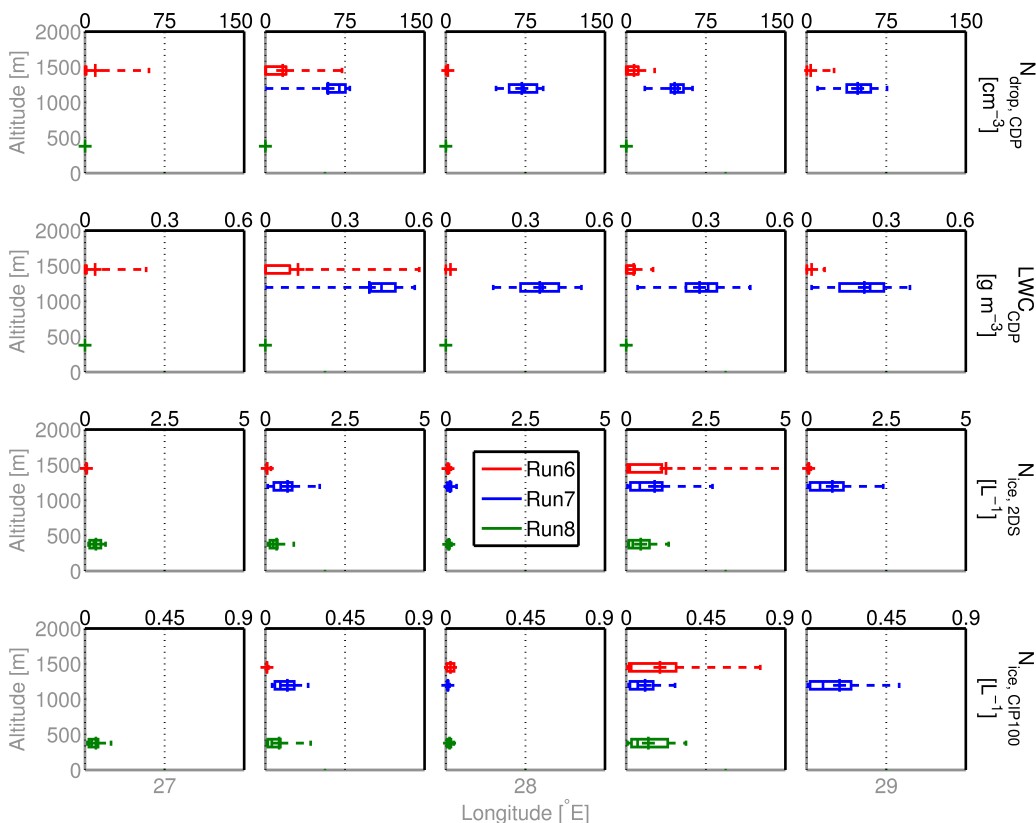

**Figure 10.** Percentile plots of CDP droplet number concentration (**top row**), CDP liquid-water content (**second row**), 2DS ice number concentration (**third row**) and CIP100 ice number concentration (**bottom row**) measured over the ocean. As in Fig. 7, columns represent different longitude bins and data are coloured by SLR (as shown in the legend).

## 4.2 Aerosol

Aerosol number concentrations measured by the various probes on board the aircraft are reported in Table 5. CPC particle concentrations are greater at high altitudes ($>800$ m) over both surfaces, as is the total concentration recorded by the PCASP over sea ice. The gradient in CPC concentration is greatest over the ocean, with a high altitude measurement of over $4\times$ that measured at low altitude. These number concentrations are not observed to the same extent in the PCASP data.

Over the ocean, the concentrations measured by the PCASP (both total and $>0.5\,\mu$m) and CAS-DPOL are approximately constant with altitude. Concentrations of large aerosol (which may act as ice nucleating particles, $D_P > 0.5\,\mu$m) are approximately uniform with altitude over the ocean. Over the sea ice, a greater loading of coarse-mode aerosol is measured nearer the surface and this consistency with altitude is not observed.

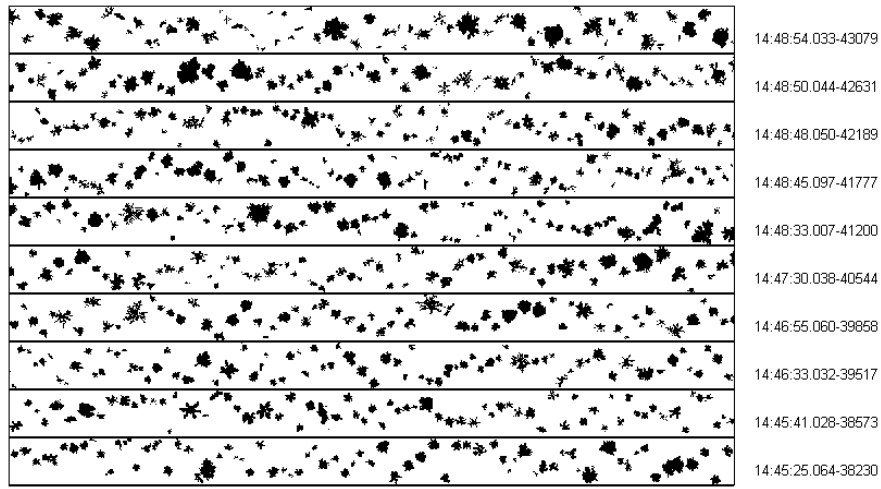

14:48:54.033-43079
14:48:50.044-42631
14:48:48.050-42189
14:48:45.097-41777
14:48:33.007-41200
14:47:30.038-40544
14:46:55.060-39858
14:46:33.032-39517
14:45:41.028-38573
14:45:25.064-38230

**Figure 11.** Example CIP100 data from run 8. Vertical width of image strip represents a size range of 6.4 mm.

**Table 5.** Background aerosol information, split into high ($>$800 m) and low altitude ($<$800 m) data over the respective surfaces. Arithmetic mean values of number concentration (cm$^{-3}$) are reported, with one standard deviation in brackets.

| Instrument | High altitude over ice | Low altitude over ice | High altitude over sea | Low altitude over sea |
|---|---|---|---|---|
| CPC | 351.0 (410.2) | 133.4 (34.0) | 595.0 (836.8) | 129.0 (68.2) |
| PCASP | 109.4 (57.2) | 86.2 (21.1) | 41.2 (31.4) | 48.3 (22.0) |
| PCASP ($>$0.5 μm) | 0.95 (4.76) | 1.94 (4.97) | 0.17 (1.62) | 0.54 (3.85) |
| CAS-DPOL | 1.27 (3.72) | 11.2 (19.7) | 2.48 (7.82) | 2.27 (7.29) |

In general, the number concentrations measured over the ocean are lower than over the sea ice. Figure 14 displays the size distributions from the PCASP and CAS-DPOL split into high and low altitude data. Small particles measured by the PCASP ($0.1\,\mu m < D_P < 0.5\,\mu m$) reach a greater concentration over the sea ice than over the ocean. Low altitude CAS-DPOL concentrations over the sea ice are heightened with comparison to the high altitude data; it is possible that swollen aerosol particles associated with the haze layer were being measured, enhancing the number concentration. Such particles may not be removed from these data using the CDP LWC $\leq$0.01 g m$^{-3}$ out-of-cloud threshold applied here. Over ocean, data from both instruments are more comparable; however, the low altitude PCASP data show a greater loading across most sizes.

Non-refractory sub-micron aerosol composition measured by the AMS is shown in Fig. 15. Technical issues prevented continuous measurement over the ocean, with problems occurring during run 7 especially. The measured nitrate mass loading remains low and consistent throughout. The sulphate loading is variable with altitude, especially over the transition region between ice and ocean. Higher mass loadings are measured at higher altitudes ($\sim$1 μg m$^{-3}$ at 1400 m): increasing during the

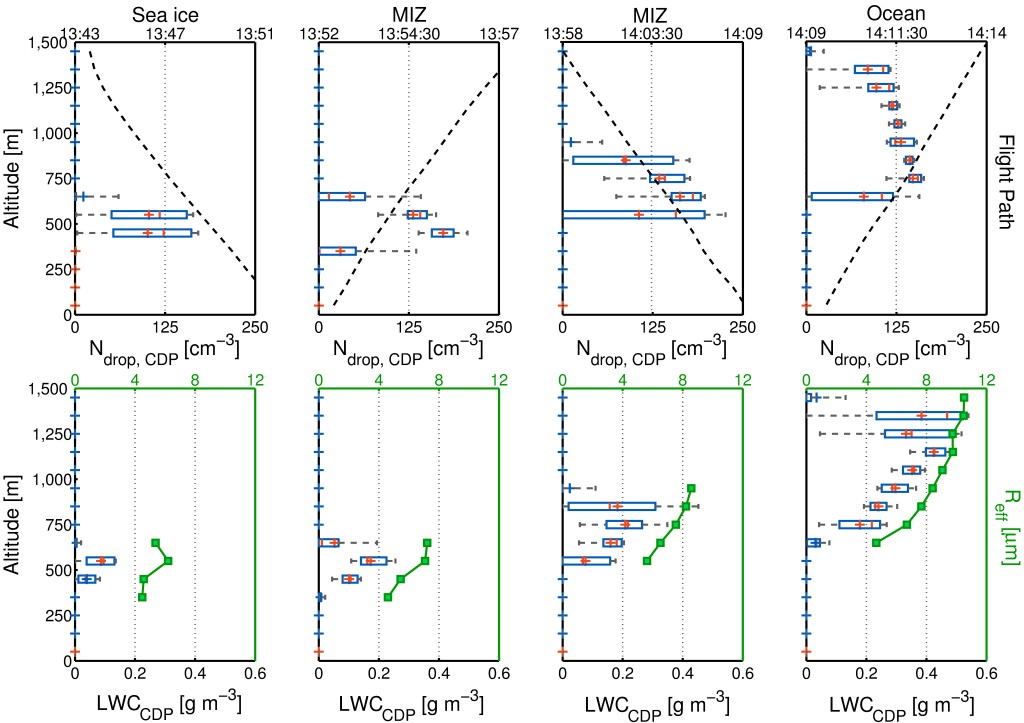

**Figure 12.** CDP data from the sawtooth profiles. Cloud droplet number concentration (cm$^{-3}$, **top row**) and derived LWC (g m$^{-3}$, **bottom row**) are shown. Box edges again indicate the 25 % and 75 % thresholds of the data, mean values are shown as a red cross and outliers extend to the 10 % and 90 % thresholds of the data. The altitude of the aircraft is indicated (black, dashed) in the top row and mean droplet effective radius – in μm, derived from CDP measurements – is shown (green) in the bottom row. Columns indicate the different profiles, transitioning from sea ice to ocean from left to right.

last SLR over the sea ice (run 5), the signal becomes highly variable over the sawtooth profile. Such variability is also observed in the organic and ammonium traces. PCASP particle number and SP2 black carbon (BC) mass observations follow the same trends throughout the science period, both mirror the same sinusoidal pattern over the MIZ. Both signals are variable, with increases observed at high altitudes, but no distinct differences are observed between sea ice and ocean measurements.

## 4.3   Boundary layer dynamics

Turbulent kinetic energy (TKE) and sensible and latent heat fluxes measured along the flight path are shown in Fig. 16. Approximate MetUM ice fraction is shown in the left-hand and middle columns. Over the sea ice, both the sensible and latent heat fluxes and the TKE remain relatively constant at about 0 W m$^{-2}$ and 0.5 m$^2$ s$^{-2}$ respectively. More variability is observed in these three parameters over the ocean. Sensible heat fluxes range from -20 to 0 W m$^{-2}$ at low altitude over the sea ice, whilst substantially greater values of >30 W m$^{-2}$ are observed over the ocean, with >100 W m$^{-2}$ measured in some instances. A similar difference is observed with the latent heat fluxes with variable measurements of approximately 50 W m$^{-2}$ over the

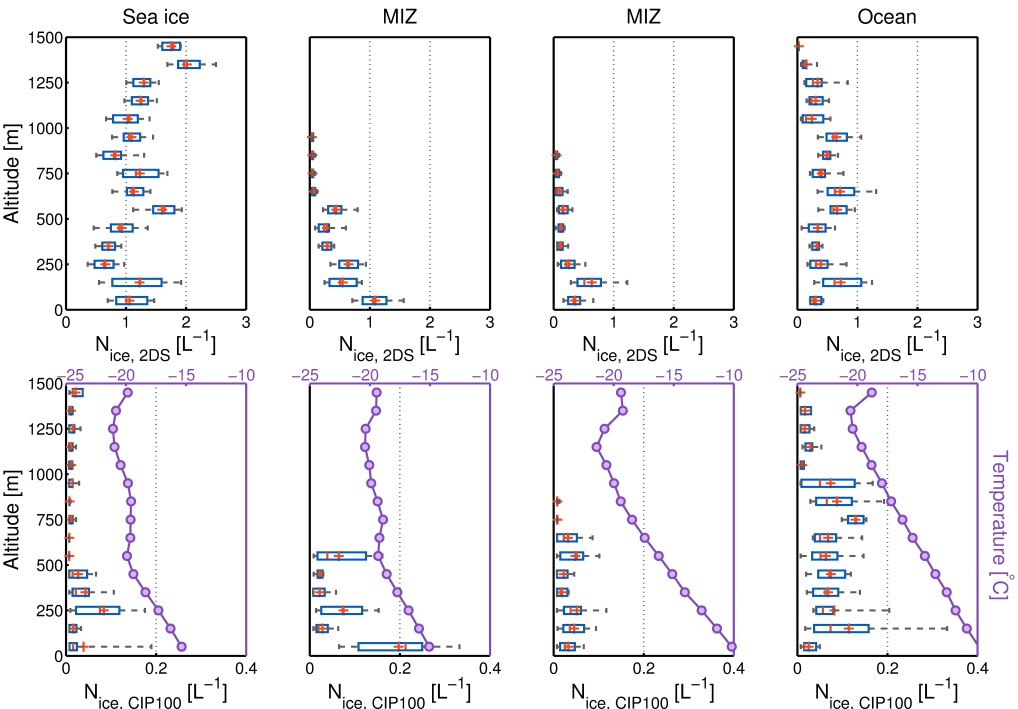

**Figure 13.** Ice number concentrations $(L^{-1})$ from the 2DS and CIP100 from the sawtooth profiles over the transition region. Data are displayed similarly to Fig.12. Temperature is overlaid (purple) in the bottom row.

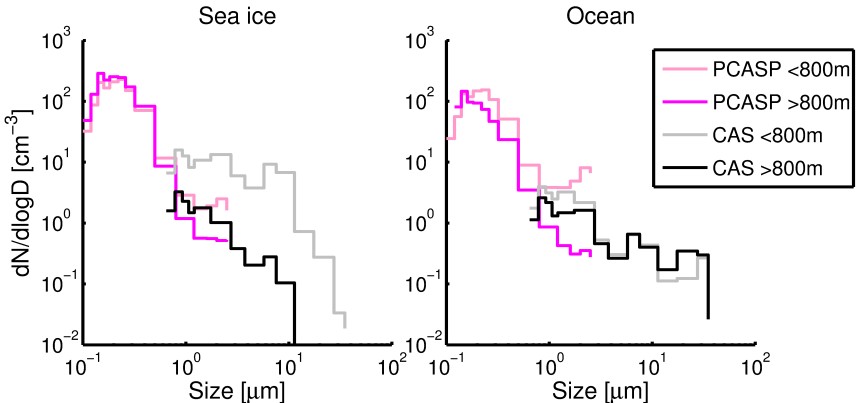

**Figure 14.** PCASP and CAS-DPOL particle size distributions over sea ice and ocean. Data are split into lower and higher than 800 m to reflect altitude influences. Only out of cloud (CDP LWC $\leq$ 0.01 g m$^{-3}$) data are included.

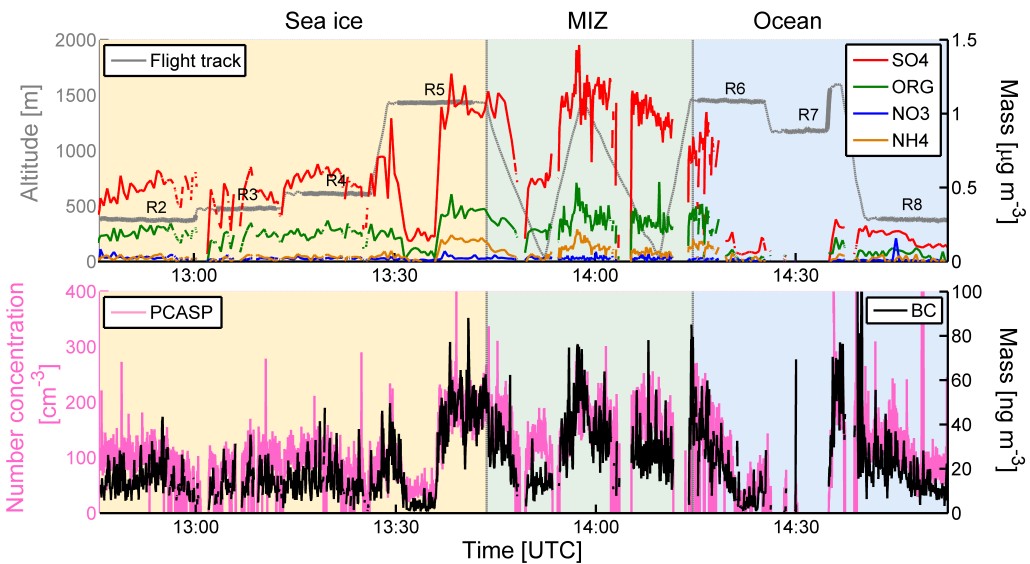

**Figure 15.** AMS and SP2 mass loading and PCASP number concentration time series. **Top row:** Flight track is shown in grey (with SLRs indicated in bold) with AMS species indicated by the legend in the top right. **Bottom row:** Aerosol number concentration and black carbon mass loading from the PCASP (pink) and SP2 (black) respectively. Only out of cloud (CDP LWC$\leq$0.01 g m$^{-3}$) data are included.

ocean, contrasting observations of $\sim$0 W m$^{-2}$ over the sea ice. Low altitude ($\sim$350 m) TKE increases from approximately 0-0.5 m$^2$ s$^{-2}$ to 1.5 m$^2$ s$^{-2}$ over the transition. TKE, sensible heat fluxes and latent heat fluxes all increase and become more variable over the MIZ and ocean with comparison to the sea ice, with the greatest values typically observed at low altitude over the ocean.

5     The turbulence and AIMMS probes recorded vertical velocity throughout the science period. The turbulence probe suffered some icing effects during runs 7 and 8, whilst the AIMMS probe collected no data for run 8 due to a technical issue. Averaged PDFs from over the sea ice and ocean are shown in Fig. 17. The turbulence probe and AIMMS PDFs compare well. The sea ice PDF displays little variation, with the majority of measurements lying close to the mean value. Maxima and minima of the distribution are approximately $\pm$1 ms$^{-1}$. In comparison, the ocean PDFs are significantly broader, with more variability

10    from the mean observed. Maxima and minima of the ocean PDF extend to $\pm$2 ms$^{-1}$ and the distribution is skewed toward updraughts.

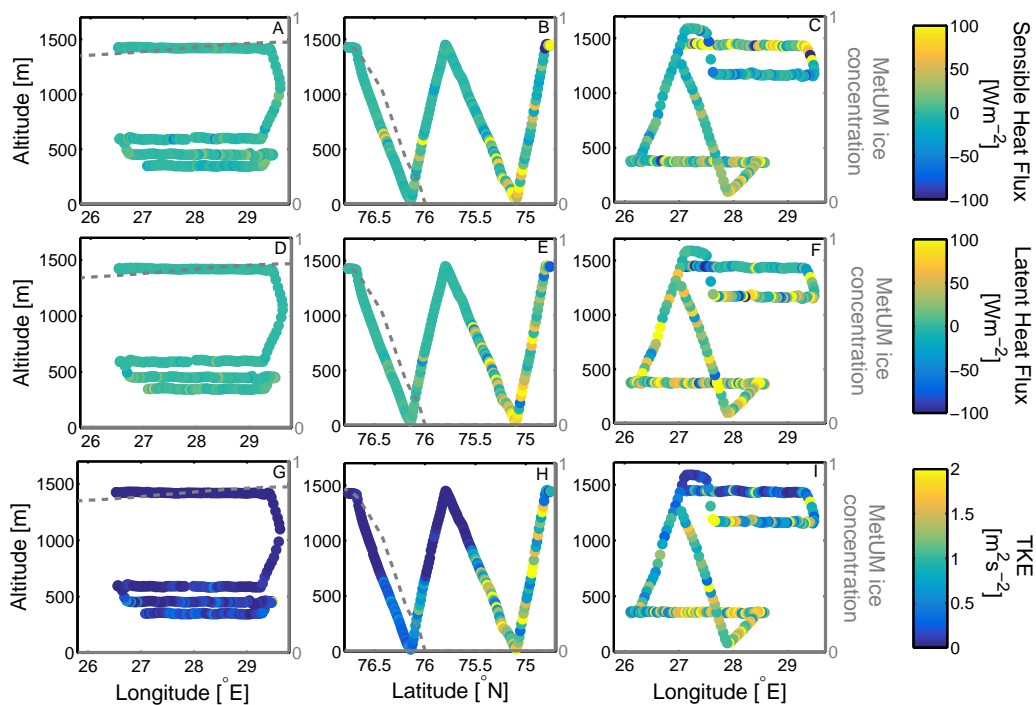

**Figure 16.** Sensible (**top row**) and latent (**middle row**) heat fluxes and turbulent kinetic energy (TKE, **bottom row**) calculated at 10 s intervals along the flight path. The path of the aircraft with respect to latitude (**middle column**) or longitude (**left- and right-hand columns**) is shown, with the measurements indicated in colour. The left column displays data from over the sea ice, whilst the middle and right columns show MIZ and ocean data respectively. Approximate MetUM sea ice fraction is shown (grey, dashed) over the sea ice and MIZ (**left** and **middle columns**), and is absent over the ocean (**right column**).

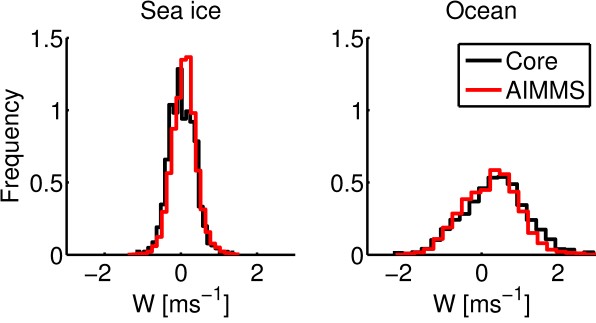

**Figure 17.** Probability density function (PDF) of updraught velocity from the core turbulence (black) and AIMMS (red) probes. Data from runs 2, 3 and 4 are used for the sea ice, and data from runs 6 and 7 for the ocean. Data from each SLR are normalised such that the mean value is zero.

## 5 Discussion

### 5.1 Sea ice

The averaged data over sea ice point toward a low-altitude cloud with a low liquid water content (Fig. 6). Ice crystal concentrations are spatially variable within the cloud (Fig. 7) yet they are consistent, suggesting only primary ice nucleation was active. The temperature within the cloud was between -18°C and -20°C (Fig. 6); far below the range required for secondary ice production (e.g. Hallett and Mossop, 1974).

Data from runs 2, 3 and 4 depict the typical structure of a single-layer Arctic mixed-phase cloud (e.g. Verlinde et al., 2007; Morrison et al., 2012; Vihma et al., 2014): a liquid-layer at cloud top, with ice formation and aggregation below. Such processes are inferred from the relative quantities of ice crystals measured by the 2DS and CIP100 instruments (Fig. 7), as the latter can measure much larger ice crystals than the former. In Fig. 6**C**, the ice mode is smaller at cloud top (panel 6**C**(c), run 4), than cloud base (panel 6**C**(a), run 2). This indicates that ice may be being nucleated towards cloud top; however, this cannot be verified with these data and vertical mixing likely has an influence.

Run 5 was planned to be an above cloud aerosol run; however, the 2DS and CIP100 instruments detect notable ice concentrations to the western end of the run (Fig. 7). This suggests that precipitation from above was being sampled, which could possibly be from the high altitude cirrus layer observed closer to Spitsbergen (Figs. 2 and 4). Additionally, RH data from the high-latitude dropsondes (Fig. 3**C**) indicate the possibility of a higher cloud layer (∼2000 m – 3000 m) in this region. CPI ice crystal images during run 5 (Fig. 8) also point towards the sampling of another cloud, as the imaged bullet rosettes typically form at higher, colder altitudes and aggregate as they descend. This ice precipitation was observed predominantly to the west of and was separate to the main cloud layer sampled over the sea ice, and thus was excluded from the comparison with the ocean cloud layer.

Observed aerosol concentrations varied substantially with altitude over the sea ice. Results show elevated sulphate, PCASP, CPC, and BC measurements during run 5 (Table 5 and Fig. 15); the latter of which is consistent with the Asian BC plumes identified during the ACCACIA campaign (Liu et al., 2015). These plumes contained an average BC mass loading of ∼27 ng s m$^{-3}$ during flight B762, consistent with the low altitude measurements here (Fig. 15). Similarly, this mass loading is also equivalent to the annual median BC concentration measured at Zeppelin, Svalbard over the period 1998-2007 (Eleftheriadis et al., 2009). PCASP data do not increase as much as the CPC data at high altitudes, suggesting either a pollution layer characterised by small particle sizes (3 nm < $D_P$ < 0.1 μm) or new particle formation at these heights.

More large particles are measured close to the sea ice surface, as shown by the >0.5 μm PCASP and CAS-DPOL measurements (Table 5). This suggests contributions of primary particle emissions from the surface. A large fraction of these low altitude particles were sea salt (Young et al., 2016), which could have been transported from the nearby ocean or lofted into the atmosphere by near-surface winds over polynyas or leads in the sea ice. Additionally, frost flowers could be a possible surface source of modified sea-salt aerosol (Xu et al., 2013); however, their characteristic signature would not be detectable by the analysis presented by Young et al. (2016).

Observed ice crystal concentrations vary little over the sea ice, with slight increases toward cloud base attributable to aggregation and precipitation out of the cloud. Using the D10 parameterisation, predicted INP concentrations were $0.9\,L^{-1}$, $1.90\,L^{-1}$ and $3.31\,L^{-1}$ for aerosol concentrations measured by the aircraft filters (Young et al., 2016), PCASP, and CAS-DPOL respectively (listed in Table 3). When using D10, the filter data produce best agreement with the mean 2DS ice concentration in the mixed cloud layer. D10 predictions using the probe data overestimate with comparison to the mean ice concentrations observed in the cloud; an overestimation which can be explained by incorrectly assuming that all predicted INPs nucleate to form ice crystals. The large fraction of coarse-mode sea salt particles identified over the sea ice (Young et al., 2016) would be unlikely to contribute to the INP population as these are inefficient INPs. Despite this, given the uncertainties in the parameterisation itself, these predictions do well to replicate the ice observed in the cloud. T13 was derived from forest ecosystem data, and therefore represents an environment with plentiful biological aerosol particles that may act as INPs. T13 predictions are highly variable for these data, giving $0.07\,L^{-1}$ with the filter data and $19.7\,L^{-1}$ with the CAS-DPOL. This variability is not surprising, as small increases in an INP-active aerosol population – such as that used to derive T13 – would cause significant changes in the ice crystal number concentrations in the clouds. Despite this, the T13 parameterisation agrees well with the observed ice number concentrations when PCASP data is used for its evaluation ($1.10\,L^{-1}$).

For comparison, parameterisations based on mineral dust data were considered. N12 and D15 were evaluated with the dust size distribution and number concentration of dust particles $>0.5\,\mu m$, derived from the aircraft filters, respectively (Table 3). As discussed by Young et al. (2016), this dust loading is likely under-represented as a result of the analysis technique and collection efficiency issues. Evaluations of N12 with measured, double, and triple dust loadings all compare well with the run 2 ice data, illustrating a lack of sensitivity to this input. However, N12 agrees best with observations when the measured dust concentrations from the filters are used. Agreement with observations is poor when applying the D15 parameterisation. This parameterisation was developed to simulate the high nucleating efficiencies of mineral dusts at temperatures $<$-20 °C; therefore, our data are at the upper limit of its applicability. Given the good agreement of N12 with our observations, at temperatures well within the range represented by the parameterisation, it can be speculated that the ice number concentrations observed over the sea ice may be explained by the dust loadings present.

Over the sea ice, the liquid-water content is low ($\sim$0.03 g m$^{-3}$, Fig. 6A) and the mean droplet radius is small ($\sim$4-5 µm, Fig. 12), with mean droplet number concentrations of $110 \pm 36$ cm$^{-3}$. It is possible that the cloud layer interacted with aerosol from above, via entrainment processes, as suggested by the presence of numerous small cloud droplets; with more CCN available, more cloud droplets can form (Jackson et al., 2012). Mean droplet number concentrations from the CDP (Fig. 6) are consistent with both the high and low altitude PCASP number concentrations measured (Table 5), further suggesting that the ambient aerosol mixed with the cloud layer from above and below.

The vertical velocity PDFs (Fig. 17) and TKE (Fig. 16) suggest that the boundary layer over the sea ice is stable with relatively little mixing compared to downstream. Little variability and low values are observed in the measured TKE, sensible heat flux, and latent heat flux (Fig. 16). The potential temperature profiles from the dropsondes (Fig. 3) also indicate weakly stable stratification with very strong temperature inversions, emphasising that boundary layer mixing was inhibited over the sea ice. A lack of substantial vertical air motions may explain the low LWC; by the WBF mechanism, the ice crystals act as a

sink for vapour in the cold temperatures observed ($\sim$-20 °C). This causes the suppression of the liquid phase – via a suppressed supersaturation – and cloud droplets remain small and fewer in number.

## 5.2 Marginal ice zone

When transitioning from sea ice to ocean, both cloud depth and cloud base height increase (Fig. 12). The most significant change in cloud microphysics over the transition is in the liquid phase, where the liquid water content and mean droplet size increase from $0.1\,\mathrm{g\,m^{-3}}$ to $0.4\,\mathrm{g\,m^{-3}}$ and $5\,\mu\mathrm{m}$ to $10\,\mu\mathrm{m}$ respectively. Mean $N_{drop}$ peaks at $145 \pm 54\,\mathrm{cm^{-3}}$ during profile 7, and falls to $120 \pm 33\,\mathrm{cm^{-3}}$ during profile 8. This decrease in $N_{drop}$ is accompanied by an increase in $R_{eff}$ towards cloud top, from $8\,\mu\mathrm{m}$ to $10\,\mu\mathrm{m}$, indicating enhanced collision coalescence within the deeper, ocean-based cloud layer (Fig. 12).

A high $N_{drop}$ over the sea ice would reflect incident solar radiation efficiently (Twomey, 1974); however, as the sea ice surface itself is highly reflective, the net impact of the cloud on the sea ice radiative interactions is difficult to interpret. As the sea ice gives way to the MIZ, the increased droplet number concentration, mean droplet effective radius, liquid-water content, and cloud depth suggest that the cloud optical thickness and albedo also increase (Twomey, 1974). With increased optical and geometrical thickness, upwelling LW radiation from the surface can be trapped and incident solar radiation can be scattered more efficiently. Radiative predictions are hindered by the weak solar heating experienced at the surface during the Arctic spring; therefore, these properties would likely cause a net warming at the surface due to the increasing LW influence from the ocean, and could potentially enhance the melting processes of the nearby sea ice (Palm et al., 2010). Over the ocean, droplet number concentrations decrease throughout the cloud, whilst the mean droplet effective radii increase. With fewer, larger, cloud droplets, the cloud over the ocean may not be as efficient as scattering solar radiation as the sea ice cloud, if their microphysical properties are compared irrespective of their environments. However, the thick cloud over the ocean would act to significantly increase the net albedo of the ocean regime, whilst the net impact of the sea ice cloud is unclear.

The ambient temperatures experienced within the cloud layers (Fig. 13) remain colder than required for secondary ice production and warmer than the homogeneous freezing threshold throughout. The observed 2DS ice crystal concentrations are consistently low – approximately $0.5\text{-}1.5\,\mathrm{L^{-1}}$ – throughout the transition, again indicating that only primary ice nucleation was active. Higher ice concentrations are observed at high altitudes (up to $2\,\mathrm{L^{-1}}$ at $\sim$1350 m) over the sea ice, suggesting that some precipitation was again measured from a higher cloud layer. This agrees with the conclusion from run 5, as both this profile and run 5 were conducted close together (1400 m) and to the west ($\sim$26.5-27 °E) of the main science region. Though some seeding from above is probable in both run and profile 5, the ice concentrations measured during the SLRs over sea ice and ocean – and the remaining profiles over the MIZ – suggest that this is not the case for the majority of these data.

The sulphate, organic, ammonium, and BC loadings vary almost sinusoidally with altitude over the transition region, with maxima reached at the peaks of the sawtooth profile (Fig. 15). These peaks occur above the cloud layer, as indicated in Fig. 12. These species commonly act as CCN in the atmosphere and could be acting to increase the cloud droplet number concentration across the MIZ. There is no evidence that the organic species influences the ice phase, as the former varies significantly whereas the latter remains approximately constant. The observed increases in mass loading are small, as are the increases in number

concentration measured by the PCASP (Fig. 15), therefore it is unlikely that they are the cause of the significant microphysical changes observed.

Measured surface heat fluxes – both sensible and latent – become more variable and increase over the transition to ocean, with greatest values measured at low altitudes (<1000 m). The approximate MetUM ice fraction indicated in Fig. 16 mirrors the transition of this variability, with little deviation from $0\,\mathrm{W\,m^{-2}}$ measured when sea ice is present and significant variability developing over the transition to open water. Similar changes are observed in the TKE data, where increased turbulence is induced as the air mass moves over the broken sea ice and comparatively warm ocean.

## 5.3 Ocean

Between sea ice and ocean, the most prominent microphysical difference is in the liquid phase. The observed cloud is deeper, with a mean LWC and droplet number concentration of $\sim$0.4 g m$^{-3}$ and $63 \pm 30$ cm$^{-3}$, respectively (Fig. 9).

Observed ice crystal number concentrations within the oceanic cloud are similar to those measured over the sea ice (0.5-1 L$^{-1}$) and the D10 INP predictions are in reasonable agreement ($\sim$1-2 L$^{-1}$, Table 3). T13 predictions are more variable with the input data, resulting in 2.66 L$^{-1}$ and 0.34 L$^{-1}$ using PCASP and CAS-DPOL data respectively. As discussed previously, T13 is particularly sensitive to the number concentration of aerosol particles used as input, with little difference in N$_{\mathrm{aerosol}}$ causing significantly different predictions (Table 3). The CAS-DPOL prediction is in better agreement with the concentrations observed in this case than the PCASP prediction. The ice phase measurements are consistent with altitude; however, there is an increase at the base of the mixed cloud layer ($\sim$700 – 900 m). This increase could be due to sedimentation of ice crystals or low sampling statistics at these altitudes. However, the presence of dendritic ice crystals (Fig. S2) combined with an ice concentration of $\sim$5 L$^{-1}$, a mean temperature of approximately -16 °C, and an enhanced ice crystal mode in the size distributed data (Fig. 9C(d)) suggests that ice-ice collisions may have taken place at cloud base (Rangno and Hobbs, 2001). Dendritic crystals are susceptible to break up, and have been shown to fragment due to air velocity alone (Griggs and Choularton, 1986). The ambient temperatures were too low to suggest secondary ice formation via the Hallet-Mossop pathway (Hallett and Mossop, 1974), but it is possible that some crystal fragmentation due to collisions enhanced the observed primary ice concentration at these altitudes.

Probe icing was an issue during run 7 and this effect can be seen in the sharp increase in the CIP100 and corresponding decrease in the 2DS ice number concentrations in Fig. 9C(b). Run 8 was intended to be a low altitude, below cloud run; however, precipitation particles were again observed. Contrasting run 5, this precipitation is related to the observed ocean cloud layer sampled above during runs 6 and 7: images from the CIP100 (Fig. 11) indicate that these particles are large and rimed, inferring an interaction with liquid droplets within the mixed layer above.

Aerosol data were not available for a significant fraction of the ocean component of the flight. Despite this, it can still be seen that the mass loadings of all AMS-measured species are low in this region (Fig. 15). The measured BC loading is more variable over the ocean than the sea ice or MIZ, varying from approximately 10 ng m$^{-3}$ to 100 ng m$^{-3}$. This variability is mirrored by the particle number concentration measured by the PCASP. The heightened BC loadings are consistent with the monthly average reported by Eleftheriadis et al. (2009) for Feb/Mar at Zeppelin station, Svalbard ($\sim$80 ng m$^{-3}$).

There is also consistency between the high and low altitude measurements from the CAS-DPOL and PCASP ($>0.5\,\mu m$) over the ocean, suggesting a constant vertical profile of large aerosol in this region. Aerosol number concentrations were found to be lower in general over the ocean than over the sea ice. Therefore, there is a lack of evidence for a significant surface flux of aerosol particles over the transition to the ocean that may have affected the cloud microphysics. No aircraft filters were exposed over the ocean; therefore, the composition of this coarse-mode aerosol could not be established.

From Fig. 3, the atmosphere is notably warmer over the ocean than over the sea ice. There is a steady increase in the boundary layer potential temperature measured by the dropsondes from north to south. The changes in the $\Theta$ profiles are most prominent in the boundary layer, with less variability observed $>1500\,m$. The near-surface temperature inferred by the dropsondes increases by $13\,^\circ C$ between the most northerly and southerly latitudes sampled. Over the transition from sea ice to ocean, the inferred temperature difference at the surface was approximately $6\,^\circ C$. The ocean surface was therefore significantly warmer than the sea ice, and this sharp temperature gradient affected the both structure of the boundary layer and any clouds that formed within it.

The broader vertical wind PDFs over the ocean (Fig. 17) suggest increased turbulence levels and mixing within the boundary layer. This is in agreement with Figs. 3 and 16: the surface temperature increased with transition from the frozen sea ice to the warm ocean, thus increasing sensible heat and latent heat fluxes from the surface. At low altitude over the ocean, both of these measures routinely exceeded $20\,W\,m^{-2}$, whilst they remained consistent at approximately $0\,W\,m^{-2}$ over the sea ice. Measured TKE was significantly higher at low altitude over the ocean than over the sea ice, suggesting a much more turbulent boundary layer over the open water. The contrast between the observed cloud microphysics over sea ice and ocean is most likely due to these increased fluxes and induced turbulent motions. The warmer, more turbulent boundary layer promotes the efficient collision-coalescence (and subsequent growth via sustained supersaturations) of cloud droplets, promoting a higher mean droplet effective radius and lower number concentration (Fig. 12). A consistent source of heat and moisture to the BL, enhanced turbulence (Fig. 16), a deeper cloud layer, and a greater liquid water content (Fig. 9) allowed rimed snowflakes to form which precipitated from the cloud. This precipitation will act to deplete the liquid in the cloud, potentially leading to cloud break up further downstream.

## 5.4 Study comparison

The sea ice cloud has little in common with the single-layer statocumulus case observed during M-PACE (Verlinde et al., 2007). Conversely, the microphysics observed here agrees better with observations made during the M-PACE cirrus case study: the high-altitude, predominantly-liquid, cloud layer observed below the cirrus cloud has a similar LWC to that measured here. Low ice number concentrations were identified in this layer, as it was found to be connected to the cirrus cloud above it through precipitating snow. In addition, this cloud layer was observed at a similar temperature (approximately -20°C) as the sea ice cloud observed here. The close proximity to the cold sea ice surface could be causing this cloud to behave like a high altitude, decoupled cloud: with little-to-no surface sensible and latent heat fluxes affecting the cloud (with little variability from $0\,W\,m^{-2}$ observed, Fig. 16), the resultant microphysics may evolve as it would higher up in the troposphere. During ISDAC,

the springtime single-layer MPS observed over the sea ice had a similar mean LWC to our sea ice cloud ($0.1 \pm 0.13\,\mathrm{g\,m^{-3}}$, Jackson et al., 2012), further suggesting this is a common observation of MPS over sea ice.

Our measurements show that the cloud downstream over the ocean was deeper than the sea ice cloud, agreeing with the observations of Palm et al. (2010). In contrast to the sea ice case, our ocean observations compare well with the ocean-based, single-layer stratocumulus observed during M-PACE (Verlinde et al., 2007; Jackson et al., 2012). Low droplet concentrations ($46 \pm 30\,\mathrm{cm^{-3}}$) and a comparable mean LWC ($0.19 \pm 0.12\,\mathrm{g\,m^{-3}}$) were measured during M-PACE (Jackson et al., 2012) and, coupled with our observations, this suggests that such properties are common amongst ocean-based single-layer MPS. Through a comparison between M-PACE and ISDAC data, Jackson et al. (2012) also concluded that this larger mean LWC during M-PACE was caused by moisture fluxes from the ocean below.

Mean droplet number concentrations varied from $110 \pm 36\,\mathrm{cm^{-3}}$ over the sea ice, to $145 \pm 54\,\mathrm{cm^{-3}}$ over the MIZ, to $63 \pm 30\,\mathrm{cm^{-3}}$ over the ocean. These concentrations were variable with altitude (Figs. 6, 9, 12), and also varied substantially with longitude over the sea ice (Fig. 7) and ocean (Fig. 10) where such data was available. These values are consistent with the first ACCACIA spring case reported by Lloyd et al. (2015) but not the second: as also concluded by the authors, their second spring case was subjected to a higher aerosol loading which enhanced the droplet number concentration of the cloud. Consistency between the liquid phase in this study; spring case 1 from Lloyd et al. (2015); and the MPS observations reported by other Arctic studies (e.g. Verlinde et al., 2007; McFarquhar et al., 2011; Jackson et al., 2012) suggests that droplet number concentrations of $\sim 150\,\mathrm{cm^{-3}}$ or below are common amongst Arctic mixed-phase single-layer stratocumulus in the transition seasons. Mean droplet effective radii over the sea ice are comparable to previous springtime Arctic studies (e.g. $5.7\,\mathrm{\mu m}$, Earle et al., 2011), whilst the larger effective radii measured over the ocean agree better with observations of autumnal Arctic single- and multi-layer clouds (e.g. approximately $10\,\mathrm{\mu m}$ and $8$–$13\,\mathrm{\mu m}$ respectively, Klein et al., 2009; Morrison et al., 2009). These observations again suggest that larger droplet sizes may be a common occurrence in ocean-based clouds, whilst small droplets are common in clouds over sea ice, regardless of season.

The ice phase is approximately constant across the transition from sea ice to ocean. Again, these measurements agree well with the springtime ACCACIA cases presented by Lloyd et al. (2015): the ice concentrations are variable and can reach up to $\sim 10\,\mathrm{L^{-1}}$ (thought to be due to crystal fragmentation here), yet they are low on average ($0.5$-$1.5\,\mathrm{L^{-1}}$) throughout the mixed cloud layer. The sea ice cloud observed here would fall into the Type IV category established by Rangno and Hobbs (2001), as it was characterised by droplet concentrations of $>100\,\mathrm{cm^{-3}}$, small droplet effective radii, and only a few ice crystals per litre of air (Figs. 6 and 12). These findings are consistent with the classification of clouds observed during ISDAC, as discussed by Jackson et al. (2012). The ocean cloud borders on the Type V category, with larger droplet sizes, mean droplet number concentrations $<100\,\mathrm{cm^{-3}}$, and precipitation developing; however, the ice number concentrations are still in better agreement with the Type IV criteria. One could postulate that the continued development of the ocean cloud over the warm surface, with further growth of even larger cloud droplets that might subsequently freeze, could allow the cloud layer to evolve into a Type V cloud, with more ice and less liquid. The microphysical characteristics of these clouds may be more susceptible to cloud glaciation and break up via the WBF mechanism. The M-PACE clouds were categorised as Type V (e.g. McFarquhar et al., 2007; Jackson et al., 2012), with higher ice crystal (mean of $2.8\pm6.9\,\mathrm{L^{-1}}$ for mixed-phase single-layer stratus, McFarquhar

et al., 2007) and lower droplet number concentrations than reported here. From these differences, one might infer that the M-PACE clouds were simply further developed than those observed in this study, or that there may be some influence from either different geographical aerosol sources, minor secondary ice production (as suggested by Jackson et al., 2012), or seasonal dependencies. Results from ISDAC may address the geographical hypothesis, as ice crystal and cloud droplet concentrations of approximately $0.5$-$1.5 \, \mathrm{L}^{-1}$ and $\sim 150 \, \mathrm{cm}^{-3}$ were observed over broken sea ice during the early spring at Barrow, Alaska (Apr 2008 McFarquhar et al., 2011; Jackson et al., 2012). These concentrations are comparable to our sea ice observations; however, the ISDAC clouds were much warmer, with cloud top temperatures ranging from -15 °C to -12 °C. The microphysical consistency between these clouds observed in different locations may suggest that a similar source of INPs is influencing these clouds, or that the ice phase is not highly sensitive to variability in aerosol properties between different locations. Variability in droplet number between different measurement locations can be more easily explained via pollution events for example, such as that observed by Lloyd et al. (2015).

The Arctic Study of Tropospheric Aerosol and Radiation (ASTAR-2004) campaign also made cloud observations in the vicinity of Svalbard; however, much higher ice crystal concentrations (up to $50 \, \mathrm{L}^{-1}$) were observed (May 2004, Gayet et al., 2009). This phenomenon was explained by Hallet-Mossop secondary ice production. Ice enhancement due to crystal collisions was inferred at cloud base over the ocean here, but the temperature was consistently too cold to allow for secondary ice via the Hallet-Mossop pathway. The lack of dominating secondary ice in the Arctic clouds studied here is again consistent with McFarquhar et al. (2011), Jackson et al. (2012) and Lloyd et al. (2015), leading to the conclusion that the single-layer MPS present in the Arctic during early spring are typically too cold for this phenomenon, irrespective of their geographical location. Primary ice nucleation was found to be solely responsible for the ice in the clouds examined here, whilst secondary ice formation has been found to play a greater role in the late spring and summer (Gayet et al., 2009; Lloyd et al., 2015).

## 6 Conclusions

In situ aircraft observations of cloud microphysics, aerosol properties, and boundary layer structure have been presented from the Aerosol-Cloud Coupling and Climate Interactions in the Arctic (ACCACIA) campaign. Using data from one case study (flight B762, 23 March 2013, Fig. 1), we have shown how the microphysics of single-layer mixed-phase stratiform clouds can significantly change over the transition from sea ice to ocean. This study represents the first investigation of in situ, measured cloud microphysical changes over this transition, and offers insight into how the microphysics of Arctic stratiform clouds may change with decreasing sea ice extent in the future.

The conclusions of this study are as follows:

– Systematic changes in microphysical properties were observed between the sea ice and ocean, which are summarised in Fig. 18. Cloud base lifted and cloud depth increased over the transition (Figs. 6, 9, 12). Both cloud droplet number and mean size increased over the marginal ice zone (MIZ), from $110 \pm 36 \, \mathrm{cm}^{-3}$ to $145 \pm 54 \, \mathrm{cm}^{-3}$ and 5 μm to 8 μm respectively. Further downstream over the ocean, mean droplet number concentrations decreased ($63 \pm 30 \, \mathrm{cm}^{-3}$) and droplet effective radii increased (up to 10 μm) due to collision-coalescence within the deepening cloud layer. Consequently, the

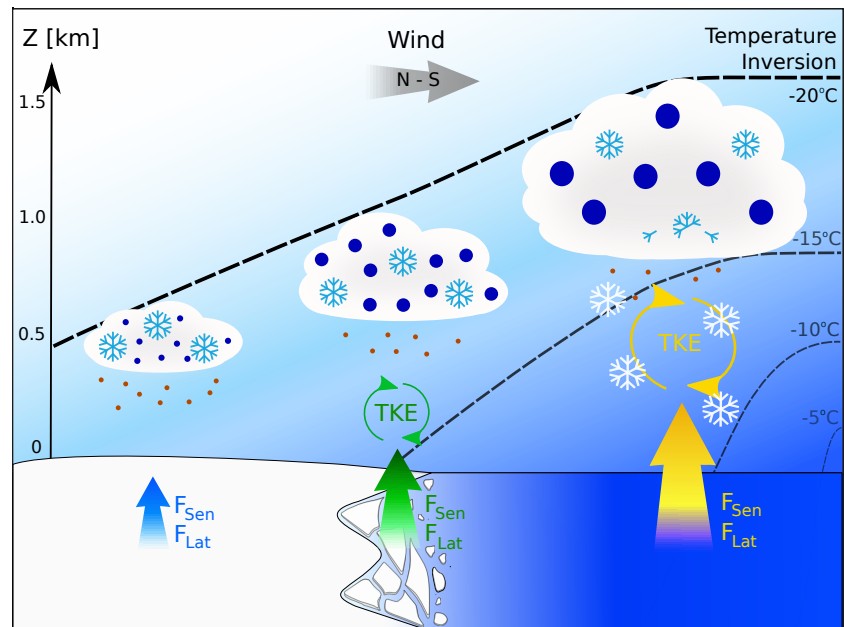

**Figure 18.** Schematic summarising the development of cloud microphysical, aerosol, and thermodynamic properties across the transition from sea ice to ocean. Aerosol particles, taken to represent the PCASP measurements, are illustrated as brown circles, and cloud droplets are similarly shown in blue. Cloud ice and snowflakes are shown as blue and white crystals respectively. The number of aerosol, droplet, and ice crystal symbols in each regime represents their number concentration in each case. Vertical arrows depict sensible and latent heat fluxes ($F_{sen}$, $F_{lat}$) from the surface, and increase in strength with size from blue, through green, to yellow. Curled arrows represent the development of turbulent kinetic energy (TKE) below the clouds and their colour and size again represent the quantities measured. Temperature isotherms illustrate the changing BL structure over the transition from the cold sea ice to the warm ocean.

liquid-water content increased almost four-fold over the full transition from sea ice to ocean. Considering the clouds alone, the clouds over the sea ice and MIZ – with relatively high numbers of small cloud droplets – would likely reflect incoming SW radiation more efficiently than the ocean cloud, promoting a cooling effect; however, as upwelling LW radiation dominates during the Arctic spring, it is more likely that each of these clouds would contribute towards a net warming at the surface by trapping upwelling LW radiation.

– The boundary layer warmed significantly from sea ice to ocean, with a near-surface temperature difference of 13 °C observed between the most northerly and southerly latitudes sampled (Fig. 3). Increased surface fluxes, vertical motion and turbulent activity (Figs. 16 and 17) infer substantially more mixing in the boundary layer over both the MIZ and ocean than over the sea ice. This is concluded to be the cause of the microphysical changes observed during this case study, as the increased heat and turbulence likely promoted the formation of more cloud droplets over the MIZ and increased the probability of efficient collision-coalescence within the deepening cloud layer over the ocean.

- The predominant change in cloud microphysics was in the liquid-phase, suggesting a similar source of INPs in both regimes. Observed ice concentrations were low and remained low over the transition (Fig. 13), suggesting only primary ice formation was active. However, evidence of crystal fragmentation was observed at cloud base over the ocean (Fig. 9), leading to minor contributions of secondary ice. The ice crystals were typically found to be larger over the ocean than over the sea ice. Such crystals were observed below cloud over the ocean as rimed snowflakes, precipitating out of the cloud (Fig. 11). Predicted ice crystal concentrations using the DeMott et al. (2010), Niemand et al. (2012), Tobo et al. (2013), and DeMott et al. (2015) parameterisations compared reasonably well, to within the uncertainty attributed to the parameterisations themselves (approximately an order of magnitude), with the ice observations over the sea ice and ocean (Figs. 6 and 9). Poorer agreement, when using the D15 parameterisation for example, could be attributed to extrapolation of the relationship to the limits of its applicability.

- Good agreement was identified between the ice crystal number concentrations measured in this study and those reported from ISDAC; both campaigns observed mean ice concentrations of approximately $0.5$-$1.5\,\mathrm{L}^{-1}$ during the early spring at different locations within the Arctic Circle. This consistency suggests that geographically-dependent aerosol sources may not have a prominent influence on the ice phase of the clouds. In contrast, substantial microphysical differences were identified between this study and previous late spring (ASTAR-2004), summer (ACCACIA) and autumnal (M-PACE) studies, emphasising that seasonality remains a crucial factor in the study of Arctic cloud microphysics.

These in situ observations offer a good test case for cloud-resolving and weather prediction model validation in the Arctic. Investigating the influence of the surface on cloud microphysics in such models, and studying how sensitive the simulated clouds are to changes in both surface and aerosol properties, could allow us to improve our understanding of how the cloud microphysics of Arctic single-layer stratiform clouds may adapt and respond to our warming climate.

*Acknowledgements.* This work was funded by the National Environment Research Council [NERC], under grant NE/I028696/1, as part of the ACCACIA campaign. G. Young was supported by a NERC PhD studentship. We would like to thank everyone involved in the ACCACIA project. G. Young would also like to thank E. Simpson for her helpful advice on schematic design. Airborne data were obtained using the BAe-146-301 Atmospheric Research Aircraft [ARA] flown by Directflight Ltd and managed by the Facility for Airborne Atmospheric Measurements [FAAM], which is a joint entity of the Natural Environment Research Council [NERC] and the Met Office. MODIS data were accessed via the NASA LAADS Web Archive. AVHRR data were produced by the NEODAAS NERC Satellite Receiving Station, Dundee University, UK (http://www.sat.dundee.ac.uk). Sea ice data were obtained from the National Snow and Ice Data Centre [NSIDC].

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
