# Peer review of "Observed microphysical changes in Arctic mixed-phase clouds when transitioning from sea ice to open ocean"

_Atmospheric Chemistry and Physics, 2016_

## Referee Comment (RC1) · Anonymous Referee #1 · 27 Jun 2016

Review of "Observed microphysical changes in Arctic mixed-phase clouds when transitioning from sea ice to ocean" by Young et al.

Recommendation: Should be acceptable for publication following minor revision This paper presents analysis from a case study showing how the cloud, aerosol, and thermodynamic properties changed during a flight of the BAe-146 over the Arctic, sampling the transition from sea ice to ocean. As it is well recognized that surface and meteorological conditions, in addition to aerosols, can impact cloud properties and that this is one of the first case studies showing the transition from sea ice to ocean during a single flight, I think that the paper adds to the body of literature about arctic clouds and should be published. Overall, the paper was well written and easy to understand

and most of the conjectures were well justified by the data. Nevertheless, I have high-lighted a couple of areas where the paper could be improved before it is accepted for final publication.

Major Comments

I would have liked to have seen more information about the meteorological conditions. This would provide a better context for the analysis of the latitudinal and longitudinal variation of cloud properties. It would be easier to attribute changes in cloud quantities in a N-S direction to variations in surface conditions if it was better known that there was no major system that could have also influenced the cloud conditions. Further, the aircraft only sampled in one direction at one particular time. Thus, when the authors stated that the 500 m cloud was not present at lower altitudes between some latitudes, this is only saying that the 500 m cloud was not present on the particular line being sampled. Although the surface conditions (i.e., open water versus sea ice) varied mostly as a function of latitude, it cannot be categorically assumed there were not variations in cloud properties (or location) as a function of longitude at the altitudes the plane was not sampling. While it is true that the measured microphysical properties did not vary a lot along the E-W runs, there is still a possibility there could be variation in the clouds or cirrus present at different altitudes in the E-W direction (e.g., the 500 m cloud layer). I think if some meteorological analysis (if available) were shown, it would better justify the analysis of cloud properties as a function of latitude.

I was a bit confused in the manner that some of the supplementary material was included in the manuscript. When I typically think of a supplement, I typically think it is related to arguments that are made outside the scope of the main paper. However, most of the supplementary figures are referred to in the body of the text. Thus, why not just make them regular figures and include them in the paper rather than including them as a supplement?

The authors stated that mixed-phase clouds in the arctic are characteristically topped

with a liquid layer which facilitates ice formation below. Although some arctic clouds definitely have this structure, I would say that the majority of arctic mixed-phase clouds do not have such a simple structure. Instead, they occur in many layers with also many layers of supercooled water sometimes embedded within them. This was discussed in Verlinde et al. (2007) and there was a study by Morrison et al. (2009) where various models characterizing such multilayer clouds were compared. This distinction should be made in the manuscript at the relevant places. There has been a greater emphasis on the single-layer clouds in the literature because they are conceptually easier to understand and study.

The authors state that their study represents the first investigation using in-situ airborne instrumentation on how cloud microphysical properties change with varying sea ice cover. I would agree that their study is the first detailed case study where such an issue was examined, with samples over varying surface characteristics on the same flight. But, I think it would be important to reference the study of Jackson et al. (2012) as they compared data from pristine and polluted conditions from two different field projects (ISDAC and M-PACE), noting that differences between the cases were caused not only by varying aerosol amounts but also by differences in the sea ice cover (open ocean versus ice). Hence, by referencing this paper, their study could be put in the proper context.

I think some details about the microphysical instrumentation are lacking. I was surprised that they did not mention any bulk measurements of liquid water content or any measures of supercooled water. Were any available? I would think the bulk measures of liquid water would be on the aircraft as a way of testing the stability of the CDP calibration. In addition, how was the phase of the particles identified? The authors note that there were mixed-phase clouds where the CDP was measuring cloud droplets and the 2DS/CIP measuring ice crystals. Was there any detailed investigation to show that the small particles were indeed droplets? McFarquhar et al. (2013) showed that in mixed-phase clouds some of the smaller particles could also be ice. Looking at the

shape of the CDP size distributions could also give some information on the likelihood that the small particles are either water or ice. Finally, was reconstruction used to extend the size ranges of either the CIP or 2DS? This should be mentioned.

The authors make the point that the ocean cloud would likely reflect incoming short-wave radiation more efficiently than a sea ice cloud. I think it is important to place this statement in an appropriate context. The sea ice will be much more efficient at reflecting the radiation than the ocean, and the role of the cloud should be considered in the context of the surface.

My final (optional) suggestion is that it would be nice to include some sort of schematic diagram in the conclusion on how the cloud, aerosol, and thermodynamic properties varied across the ocean, sea ice and transition region. I think this would be a nice way of summarizing their findings and would be a diagram that could also be well referenced in future studies.

Detailed Comments

Page 3, line 15 replace semi-colon after Crosier et al. (2015) with a comma

Page 12, line 7: I would prefer dramatically to be replaced by something more quantitative.

Page 14, line 17, is it appropriate to call this cloud cirrus if it has LWC? Maybe this comment is more directed at the Verlinde et al. (2007) paper, but I think it is also relevant for this paper.

---

## Referee Comment (RC2) · Anonymous Referee #2 · 10 Aug 2016

The manuscript reports detailled cloud microphysical studies of arctic mixed phase stratocumulus clouds probed during a single flight (B762) of the FAAM aircraft within the ACCACIA campaign in spring 2013. The authors present detailled cloud microphysical as well as aerosol and boundary layer dynamics of a transect that spans from sea ice covered arctic conditions to open ocean conditions. In contrast to earlier studies, special emphasis is put on the transition between these two regimes. As both cloud ice and aerosols have been probed, a comparison to primary ice nucleation parametrizations is made. The data presented are of high quality and rather comprehensive, they provide a clear picture of how the surface influences and sometimes dominates the cloud properties in both cases. The study enriches our understanding of arctic boundary layer clouds and provides valuable data for meteorological modelling. It remains open however, inhowfar the results can be transferred to different meteorological conditions or other seasons. From my perspective the manuscript is very well written and in most parts easy to comprehend. It may be published after considering the following single suggestion for improvement, which concerns the comparison to ice nucleation parametrisations. First I suggest that the authors include more recent ice nucleation parametrizations such as "deMott et al. ACP 15, 393, 2015" or "Tobo et al., JGR Atm. 118, 10,100, 2013" Secondly I am asking for a more detailed error analysis here. Two potential issues immediately come to mind: What is the sampling height distribution of the mineral dust?. Was it sampled only at the altitudes relevant for ice formation? Is ice predominantly nucleating at the cloud top? If not, why is the cloud top temperature taken into account? Thirdly, I feel that Figure 5 is somewhat overbusy. Especially ice number concentration from the parametrization is not well placed here. It is unclear, to what axis it belongs and to what measurement data points it should be compared.

---

## Referee Comment (RC3) · Anonymous Referee #3 · 12 Aug 2016

review of Young et al, "observed microphysical changes in arctic mixed phase clouds when transitioning from sea ice to open ocean"

This paper reports an experiment where an instrumented aircraft made detailed measurements of cloud and meteorological parameters in a widespread mixed-phase stratocumulus deck, contrasting the properties over the ocean, over the sea ice, and the transition zone between the two.

I felt that the paper was very well written, clear in its aims and in presenting the data (which can be challenging when presenting a complex case study). It is a valuable contribution to the arctic mixed-phase literature. I recommend publication, and have only a few minor suggestions for changes below.

1. Abstract – you contrast the cloud droplet concentration between sea ice and ocean as 80/cc over ice vs 90/cc over ocean.
   First point – this is not consistent with section 4.11 and 4.12 where you say there are 90/cc over the sea ice and 70/cc over the ocean. Please check.
   Second point - Do you think this difference is statistically significant? There is a lot of variability in this number over each run.
2. Abstract & elsewhere. "Evidence for crystal fragmentation" – I felt this was a bit speculative. The concentrations are clearly higher in this run as you say, but you don't really show direct evidence that this is due to fragmentation – it's just an argument based on the crystal habit and the fact that we are too cold for Hallett Mossop. It would be good in the main text to bring in a bit more of the relevant literature on breakup (Griggs and Choularton etc?). The other thing that would add weight to this argument would be to show any CIP-15/CIP-100 images which look like broken arms of dendrites.
3. Section 2.1 you say "all data are expressed as ambient with no standard temperature and pressure corrections applied". It wasn't clear to me what this meant, what kind of corrections did you not apply?
4. Section 3 – you say "In this study, the MIZ is not distinctly defined" and then proceed to give a very clear definition for what you are choosing the MIZ to represent. I think you could rephrase this.
5. Section 3.1 – I was a bit puzzled by the dropsonde data. Why are the peak RH values so low, when there is clearly liquid cloud present (hence RH=100%, or very very close)? Many sonde RH sensors have a small dry bias, but in figure 3D the cloud layers seem to have a peak RH of only 80%. Could you comment on that (in the text)?
6. section 4.1.1 – "subtly detected" – could phrase this better
7. section 4.1.1 – rosettes & aggregates fallen from Ci cloud above. Very interesting – can you say whether these are likely have to survived far enough to actually precipitate into the MPS layer itself (seeder-feeder arrangement)?
8. last part of section 4.1.2 – would be good to tell the reader at this point what they should take from the figure showing the evaluation of D10
9. section 4.2 and elsewhere. You talk about a "fog layer" here, but then of the droplets as "swollen aerosol particles". I think it's important to be clear throughout the paper

what this layer is – is it haze (ie unactivated solution droplets) or is it fog (activated cloud droplets)

10. section 4.3 and 2.2 – I think it is better to use a phrase like sea ice fraction rather than "ice concentration" (which makes reader think of snow particles in the clouds)

11. section 5.1 – talk about cirrus cloud and then "the possibility of another higher cloud layer" – makes it sound like there is a cloud higher than the cirrus, when in fact you are suggesting one in-between I think. Rephrase.

---

## Author Comment (AC1) · 2 Sep 2016

**We would like to thank the reviewers for their useful comments and suggestions which have helped us to improve the manuscript.**

**Reviewers' comments**

Reviewer 1, Reviewer 2, Reviewer 3

**Authors' response is shown in black and bulleted.**

**Please note:** Some figure numbers have been changed in the updated version of the manuscript. New figure numbers are referenced in any related comments.
* * *
Recommendation: Should be acceptable for publication following minor revision. This paper presents analysis from a case study showing how the cloud, aerosol, and thermodynamic properties changed during a flight of the BAe-146 over the Arctic, sampling the transition from sea ice to ocean. As it is well recognized that surface and meteorological conditions, in addition to aerosols, can impact cloud properties and that this is one of the first case studies showing the transition from sea ice to ocean during a single flight, I think that the paper adds to the body of literature about arctic clouds and should be published. Overall, the paper was well written and easy to understand and most of the conjectures were well justified by the data. Nevertheless, I have highlighted a couple of areas where the paper could be improved before it is accepted for final publication.

The manuscript reports detailed cloud microphysical studies of arctic mixed phase stratocumulus clouds probed during a single flight (B762) of the FAAM aircraft within the ACCACIA campaign in spring 2013. The authors present detailed cloud microphysical as well as aerosol and boundary layer dynamics of a transect that spans from sea ice covered arctic conditions to open ocean conditions. In contrast to earlier studies, special emphasis is put on the transition between these two regimes. As both cloud ice and aerosols have been probed, a comparison to primary ice nucleation parametrizations is made. The data presented are of high quality and rather comprehensive, they provide a clear picture of how the surface influences and sometimes dominates the cloud properties in both cases. The study enriches our understanding of arctic boundary layer clouds and provides valuable data for meteorological modelling. It remains open however, inhowfar the results can be transferred to different meteorological conditions or other seasons.

Review of Young et al, "observed microphysical changes in arctic mixed phase clouds when transitioning from sea ice to open ocean" This paper reports an experiment where an instrumented aircraft made detailed measurements of cloud and meteorological parameters in a widespread mixed-phase stratocumulus deck, contrasting the properties over the ocean, over the sea ice, and the transition zone between the two. I felt that the paper was very well written, clear in its aims and in presenting the data (which can be challenging when presenting a complex case study). It is a valuable contribution to the arctic mixed-phase literature. I recommend publication, and have only a few minor suggestions for changes below.

**General Comments**

I would have liked to have seen more information about the meteorological conditions. This would provide a better context for the analysis of the latitudinal and longitudinal variation of cloud properties. It would be easier to attribute changes in cloud quantities in a N-S direction to variations in surface conditions if it was better known that there was no major system that could have also influenced the cloud conditions.

- No major weather systems were influencing the sampled clouds. A weak low pressure system was present to the east; however, conditions were dominated by high pressure to the west which caused the northerly flow of air off the sea ice pack. The manuscript has been updated to include this information (Section 3.1, page 7, lines 8-11).

Further, the aircraft only sampled in one direction at one particular time. Thus, when the authors stated that the 500 m cloud was not present at lower altitudes between some latitudes, this is only saying that the 500 m cloud was not present on the particular line being sampled. Although the surface conditions (i.e., open water versus sea ice) varied mostly as a function of latitude, it cannot be categorically assumed there were not variations in cloud properties (or location) as a function of longitude at the altitudes the plane was not sampling.

- Satellite imagery was examined to provide some lateral context, and we believe these in situ cloud measurements are representative. However, we agree with the reviewer's concerns, and have added to the manuscript to make these sampling biases more explicit (Section 3.1, page 6, lines 7-8; page 7, line 1).

While it is true that the measured microphysical properties did not vary a lot along the E-W runs, there is still a possibility there could be variation in the clouds or cirrus present at different altitudes in the E-W direction (e.g., the 500 m cloud layer). I think if some meteorological analysis (if available) were shown, it would better justify the analysis of cloud properties as a function of latitude.

- Meteorological summary discussed above has been included in the manuscript for better clarity (Section 3.1, page 7, lines 8-11).

I was a bit confused in the manner that some of the supplementary material was included in the manuscript. When I typically think of a supplement, I typically think it is related to arguments that are made outside the scope of the main paper. However, most of the supplementary figures are referred to in the body of the text. Thus, why not just make them regular figures and include them in the paper rather than including them as a supplement?

- Figures S2 (time series data), S3 (spatial sea ice cloud data), and S4 (spatial ocean cloud data) have been moved into the main body of the manuscript, becoming Figs.5, 7, and 10 respectively, to reflect the reviewer's concerns.

The authors stated that mixed-phase clouds in the arctic are characteristically topped with a liquid layer which facilitates ice formation below. Although some arctic clouds definitely have this structure, I would say that the majority of arctic mixed-phase clouds do not have such a simple structure. Instead, they occur in many layers with also many layers of supercooled water sometimes embedded within them. This was discussed in Verlinde et al. (2007) and there was a study by Morrison et al. (2009) where various models characterizing such multilayer clouds were compared. This distinction should be made in the manuscript at the relevant places. There has been a greater emphasis on the single-layer clouds in the literature because they are conceptually easier to understand and study.

- The authors agree with the reviewers' concerns; therefore, it has been made clearer at several points throughout the manuscript that we are considering single-layer clouds only. This is particularly emphasised in the Introduction (Sect. 1, page 2, lines 21, 23, and 24-25) and Conclusions (Sect. 6, page 19, line 16; page 20, line 32).

The authors state that their study represents the first investigation using in-situ airborne instrumentation on how cloud microphysical properties change with varying sea ice cover. I would agree that their study is the first detailed case study where such an issue was examined, with samples over varying surface characteristics on the same flight. But, I think it would be important to reference the study of Jackson et al. (2012) as they compared data from pristine and polluted conditions from two different field projects (ISDAC and M-PACE), noting that differences between the cases were caused not only by varying aerosol amounts but also by differences in the sea ice cover (open ocean versus ice). Hence, by referencing this paper, their study could be put in the proper context.

- The Jackson et al., 2012 study has been discussed more extensively in the introductory section to illustrate their findings in the context of our study. Specifically, the following has been added: "*Jackson et al., 2012 found a greater mean liquid-water content in clouds over the ocean (during M-PACE) compared with those over the sea ice (during ISDAC)*" (Sect. 1, page 3, lines 8-10). Additionally, further discussion with reference to this study has been included in Section 5.4 (page 17, lines 18-20; lines 25-29; page 18, lines 17-18; line 30).

I think some details about the microphysical instrumentation are lacking. I was surprised that they did not mention any bulk measurements of liquid water content or any measures of supercooled water. Were any available? I would think the bulk measures of liquid water would be on the aircraft as a way of testing the stability of the CDP calibration.

- Bulk measurements of liquid water were made using a Johnson-Williams probe and, when liquid water was present, the instruments agreed well. However, an issue arose with the hot-wire probe: this instrument experienced some lag in the measured liquid water when leaving cloud, and it would take some time for the signal to return to zero. Consequently, the CDP liquid water content was the more reliable measure of the two and it was solely used to demonstrate this property. This was not explicitly detailed in the manuscript; therefore, the manuscript has been updated to include a summary of this information (Sect. 2.1.3, page 4, lines 21-23).

In addition, how was the phase of the particles identified? The authors note that there were mixed-phase clouds where the CDP was measuring cloud droplets and the 2DS/CIP measuring ice crystals. Was there any detailed investigation to show that the small particles were indeed droplets? McFarquhar et al. (2013) showed that in mixed-phase clouds some of the smaller particles could also be ice. Looking at the shape of the CDP size distributions could also give some information on the likelihood that the small particles are either water or ice.

- McFarquhar et al. (2013) found that a large proportion of small particles in liquid-dominated mixed-phase clouds, in the "small ice" size range ($35\mu m < D_{max} < 60\mu m$), had a high area ratio, suggesting sphericity. The clouds considered here would be classified as liquid-dominated due to the low ice number concentrations. Therefore, any potential small ice particles would likely be spherical. Phase identification of these small particles was limited by the instruments used; 2DS and CIP images are of too low resolution for the phase of such small particles ($< 60\mu m$) to be identified by our image analysis software. No further investigation into the phase of these small

particles was undertaken and it is possible that some were ice. A CPI was active during this case study; however, due to the low number concentrations of particles measured and a small sample volume, a statistically-significant set of images was not obtained.

Figures 6C and 9C show the size distributions measured from all the microphysical instruments used and, in each distribution, the droplet mode measured by the CAS and CDP is separate and distinct from the mode at larger sizes, measured by the CIP15, 2DS, and CIP100, which was taken to be ice. We cannot conclude the presence of small ice with these data; however, the size distributions suggest that the ice, by majority, is of larger sizes than the droplets due to the well-defined secondary mode.

Despite this, work is currently being carried out into the phase identification of small particles, and will be presented in a future publication. Extra information of the phase discrimination of cloud particles has been added to the manuscript (Section 2.1.3, page 4, lines 32-33; page 5, lines 1-6).

Finally, was reconstruction used to extend the size ranges of either the CIP or 2DS? This should be mentioned.

- Reconstruction was not used to extend the size ranges of the optical array probes. This has been made clear in Section 2.1.3 (page 5, lines 2-3).

The authors make the point that the ocean cloud would likely reflect incoming shortwave radiation more efficiently than a sea ice cloud. I think it is important to place this statement in an appropriate context. The sea ice will be much more efficient at reflecting the radiation than the ocean, and the role of the cloud should be considered in the context of the surface.

- The authors agree with this concern. This was the aim of paragraph two of Section 5.2; however, we realise that the language was not as explicit as it could be. With the updated droplet number concentrations (see below/abstract), the message of this paragraph has changed. With more small droplets over the sea ice, this cloud would be more efficient at scattering solar radiation than the ocean cloud, with fewer big droplets, if the cloud microphysics alone was compared. However, the ocean cloud would act to increase the net albedo of the ocean regime, as it is still significantly more reflective than the dark ocean surface. The radiative impact of the sea ice cloud is more difficult to interpret, as both the surface and the cloud have high albedos. We have speculated that this cloud may then have a net warming effect at the sea ice surface; however, this is speculation and we cannot prove this with our data. This paragraph has been rewritten to emphasise the environmental context of the clouds (Sect. 5.2, page 14, lines 14-25).

My final (optional) suggestion is that it would be nice to include some sort of schematic diagram in the conclusion on how the cloud, aerosol, and thermodynamic properties varied across the ocean, sea ice and transition region. I think this would be a nice way of summarizing their findings and would be a diagram that could also be well referenced in future studies.

- The authors agreed with this suggestion, as such a schematic was constructed and included in the conclusions section of the article (Fig. 18).

From my perspective the manuscript is very well written and in most parts easy to comprehend. It may be published after considering the following single suggestion for

improvement, which concerns the comparison to ice nucleation parametrisations. First I suggest that the authors include more recent ice nucleation parametrizations such as "deMott et al. ACP 15, 393, 2015" or "Tobo et al., JGR Atm. 118, 10,100, 2013"

- These parameterisations have been evaluated and included in Table 3. To accommodate this extra information, Table 3 has been re-formatted. Discussion relating to these data has also been included throughout the manuscript (Sections 4.1.1, 4.1.2, 5.1, 5.3). Predictions made using the Tobo et al., 2013 parameterisation were highly variable with aerosol input, as would be expected. This parameterisation is based upon data from a biologically-active ecosystem; therefore, small increases in the INP-active aerosol population would cause significant changes in the ice crystal number concentrations in the clouds. We do not see such sensitivity here, and the ice phase in the cloud is consistent over the transition from sea ice to ocean. The DeMott et al., 2015 (D15) parameterisation could only be evaluated over the sea ice, like the Niemand et al., 2012 parameterisation, as we only had dust data below-cloud in this location. Predictions were very low using D15 (0.05 $L^{-1}$), and this was attributed to the use of this parameterisation at its upper-most temperature limit (-20$^o$C).

Secondly I am asking for a more detailed error analysis here. Two potential issues immediately come to mind: What is the sampling height distribution of the mineral dust? Was it sampled only at the altitudes relevant for ice formation?

- Filters were primarily collected below cloud during the ACCACIA campaign to address the contribution of primary aerosol sources to the clouds. The analysis of these filters are detailed in Young et al., 2016. Only one filter set was available for this case study, and it was exposed below cloud over the sea ice. Particle data derived from these filters are therefore only strictly valid at the altitude and geographical location of collection (approximately 380m, 76.8$^o$N). Mineral dust concentrations are valid for comparison with the microphysics of the sea ice cloud, due to their close proximity (sea ice cloud base 300m, with 100m-binned data). No vertical profiles of mineral dust were obtained. The manuscript has been updated to make this clearer to the reader (Sect. 4.1.1, page 8, lines 7-8).

Is ice predominantly nucleating at the cloud top?

- It is difficult to truly identify where nucleation occurs within these clouds due to turbulent motion. However, the CIP100 (Figures 7, 13) typically shows increasing ice number concentrations with decreasing altitude, suggesting that ice growth and aggregation occurs as crystals fall through the ice layer. Also, from Figure 6, the ice mode in panel C is smaller at cloud top (run 4), than cloud base (run 2). A similar trend is clear in Figure 9 (again, panel C), where more large ice is seen at cloud base (and below cloud) than at cloud top. This indicates that ice may be being nucleated towards cloud top; however, this cannot be truly verified with these data. This has been made clearer in the manuscript (Sect. 5.1, page 12, lines 10-12).

If not, why is the cloud top temperature taken into account?

- Cloud top temperature was used in evaluating the ice nucleation parameterisations to produce an upper estimate of the INP/ice crystal concentrations. Due to turbulent motions within the cloud, ambient temperatures could not be used as an indicator of INP/ice crystals at specific altitudes. Therefore, the coldest temperature attained in the cloud – the cloud top temperature – was used to give the maximum number of

predicted INP/ice crystals that we can expect in the cloud. This has been made more explicit in the manuscript (Sect. 4.1.1, page 8, lines 10-12).

Thirdly, I feel that Figure 5 is somewhat overbusy. Especially ice number concentration from the parametrization is not well placed here. It is unclear, to what axis it belongs and to what measurement data points it should be compared.

- Figures 6B and 9B have been changed to remove the parameterisation estimates; therefore, the data presented are clearer and less confusing. Similarly, we felt the estimates included in Fig. 13 may have been misleading. Evaluations of the parameterisations are now included in Table 3 only.

**Detailed Comments**

1. Abstract – you contrast the cloud droplet concentration between sea ice and ocean as 80/cc over ice vs 90/cc over ocean. First point – this is not consistent with section 4.11 and 4.12 where you say there are 90/cc over the sea ice and 70/cc over the ocean. Please check.

- The comparison listed in the abstract related an approximate of the cloud drop number concentrations throughout the sea ice and ocean clouds. It was an approximation based on the mean data in each altitude bin in Figures 6 and 9. The specific numbers discussed in the results section – the 90 cm$^{-3}$ over the sea ice and 70 cm$^{-3}$ over the ocean referred to by the reviewer – are summaries of data at specific altitudes. For example, the quoted 90 cm$^{-3}$ referred to the 400m altitude bin only.

  We realise we made an error with these data: as seen clearly in the timeseries figure (Fig.5) and transition figure (Fig. 12), droplet number concentrations visibly increase over the transition region, yet they subsequently decrease over the ocean. This decrease was previously not acknowledged in our interpretation of the data. For clarity, droplet number concentrations increase from 110 ± 36cm$^{-3}$ to 145 ± 54cm$^{-3}$ over the transition region. Our interpretation of these data, as included in the original version, is still valid. However, mean droplet concentrations decrease to 63 ± 30cm$^{-3}$ over the ocean. This decrease is accompanied by the increase in droplet size discussed before, and we believe this trend is a result of collision-coalescence within the deep, turbulent cloud layer over the ocean.

  Our conclusions are still valid and, in fact, we are able to present more information about the development of the cloud. Our conclusion that droplet number concentrations increase over the transition is still true, as seen by the sea ice and MIZ values quoted above. However, we can now speculate how the microphysics will change further downstream in a warming boundary layer: with a consistent source of heat and moisture to the BL, enhanced turbulent motions (as seen by the TKE data illustrated in Fig. 16) and a deeper cloud layer promote efficient collision-coalescence, droplet growth, and the development of rimed precipitation. This will act to deplete the liquid in the cloud, potentially leading to cloud break up further downstream. Additionally, our updated observations now agree better with the Rangno and Hobbs (2001) criteria (Type IV sea ice, Type V ocean, with the exception of the lower ice concentrations than required).

  This decrease in droplet number concentration over the ocean was not reflected in our initial quoted comparisons, and we apologise for the confusion. We have made the language clearer in the manuscript (Abstract, Sects 4.1.1, 4.1.2, 5, and 6). For clarity, the calculated average drop concentration, with one standard deviation, over

the sea ice and ocean is now presented in the abstract (110 ± 36cm$^{-3}$ over sea ice and 63 ± 30cm$^{-3}$ over ocean, increasing to 145 ± 54 over the MIZ in between) to replace the approximate values quoted before.

Second point - Do you think this difference is statistically significant? There is a lot of variability in this number over each run.

- Yes, we do believe these data are statistically significant, given the magnitude of the standard deviations attached to the updated values.

2. Abstract & elsewhere. "Evidence for crystal fragmentation" – I felt this was a bit speculative. The concentrations are clearly higher in this run as you say, but you don't really show direct evidence that this is due to fragmentation – it's just an argument based on the crystal habit and the fact that we are too cold for Hallett-Mossop. It would be good in the main text to bring in a bit more of the relevant literature on breakup (Griggs and Choularton etc?). The other thing that would add weight to this argument would be to show any CIP-15/CIP-100 images which look like broken arms of dendrites.

- Further information from the literature has been included in the manuscript as requested (Sect. 5.3). The following has been added to the manuscript: "*Dendritic crystals are susceptible to break up, and have been shown to fragment due to air velocity alone (Griggs and Choularton, 1986)*" (Sect. 5.3, page 16, lines 3-4). CIP100 images are included in the supplementary material (Fig. S2), which show broken dendrites. These dendrite arms are particularly clear in the bottom three data rows of Figure S2.

Page 3, line 15 replace semi-colon after Crosier et al. (2015) with a comma

- Addressed in manuscript.

3. Section 2.1 you say "all data are expressed as ambient with no standard temperature and pressure corrections applied". It wasn't clear to me what this meant, what kind of corrections did you not apply?

- We did not correct measured particle number concentrations to standard temperature and pressure. Such corrections are often made to make these measurements directly comparable, giving a concentration with respect to a consistent volume of air. These corrections were not applied to the presented data due to the consistent, low-altitude of the data; therefore, we felt they were not required. However, we wished to be explicit in the manuscript that they were not carried out. The language in Sect. 2.1 of the manuscript has been improved to make this clearer (Sect. 2.1, page 3, lines 26-27).

4. Section 3 – you say "In this study, the MIZ is not distinctly defined" and then proceed to give a very clear definition for what you are choosing the MIZ to represent. I think you could rephrase this.

- With comparison to sea ice studies, our definition of the marginal ice zone is approximate. However, we realise that our chosen language is conflicting without that knowledge; therefore, we have updated this in the manuscript to read as follows "*In this study, the MIZ is approximated by NSIDC ice fractions between 10% and 90%, as indicated in Fig. 1*" (Sect. 3, page 5, lines 22-23).

5. Section 3.1 – I was a bit puzzled by the dropsonde data. Why are the peak RH values so

low, when there is clearly liquid cloud present (hence RH=100%, or very very close)? Many sonde RH sensors have a small dry bias, but in figure 3D the cloud layers seem to have a peak RH of only 80%. Could you comment on that (in the text)?

- Yes, we did experience a dry bias in the sonde measurements as highlighted by the reviewer, and this effect was rather strong. We found that in-cloud measurements only registered an RH of 84% in dropsonde #11 and 88% in dropsonde #5, when close to 100% was expected. Cloud was present at these altitudes, as indicated with the lidar data. Dry biases have been observed in dropsonde data in previous studies (e.g. Ralph et al., 2005), and such systematic biases have been attributed to a slow response time at low temperatures (Poberaj et al., 2002, Miloshevich et al., 2004). These sondes are particularly prone to these response issues when descending from a dry region into a cloudy region (Wang 2005). Wang 2005 also demonstrate that this bias can be related to the age of the sensor. We do not speculate the reason for our dry bias here, but we realise that we did not adequately discuss this in the manuscript, and this information has now been included (Sect. 3.1, page 6, lines 13-30 – paragraph restructured to accommodate new information).

6. Section 4.1.1 – "subtly detected" – could phrase this better

- This has been updated in the manuscript to the following: "*The secondary cloud layer at ~1000m, indicated by the dual RH peaks in Figs. 3C and 3D, is observed, and is dominated by ice*" (Sect. 4.1.1, page 7, line 31).

7. Section 4.1.1 – rosettes & aggregates fallen from Ci cloud above. Very interesting – can you say whether these are likely have to survived far enough to actually precipitate into the MPS layer itself (seeder-feeder arrangement)?

- From the northern-most profile over the MIZ (left-hand boxes of Fig. 13), it appears that some of the ice crystals from above are reaching the low altitudes (300m-700m) of the main cloud layer. Before run 2 begins (<12:45 UTC), CPI images show some bullet-rosettes towards the west (~26.75-27.25$^o$E) at low altitudes. Similar habits can be seen in the 2DS and CIP15 imagery. These data are not included in Fig. 5 as they are measured at a time before the main science period began; however, they are included in Fig. 7. Little liquid (both as droplet number concentration and LWC) is observed alongside these bullet-rosettes (~26.5-27$^o$E, Fig. 7); however, data shown in Fig. 7 are binned over a long distance (0.5$^o$) and therefore must be interpreted with caution in this context.

  For the majority of the sea ice cloud layer sampled (27-29$^o$E, Fig. 7), rimed crystals dominate. Other habits such as columns and dendrites are also prevalent. Bullet rosettes are not observed within the cloud to the same extent as at the western fringes of this cloud; however, few are observed with the CPI. Due to the dominance of large rimed ice crystals, it is therefore difficult to conclusively state if these bullet-rosettes precipitated into the main cloud layer considered. This has been made clearer in the manuscript (Sect 4.1.1, page 7, lines 28-29; page 9, lines 28-31).

8. Last part of section 4.1.2 – would be good to tell the reader at this point what they should take from the figure showing the evaluation of D10

- The D10 points on Fig. 9 (and Fig. 6) have now been removed to avoid confusion, as this was highlighted as a potential issue by Reviewer 2. The results of the parameterisations are listed in Sect. 4.1, and comparisons between the observed data and predicted ice crystal/INP concentrations from the parameterisations are

discussed in Sects. 5.1 and 5.3.

9. Section 4.2 and elsewhere. You talk about a "fog layer" here, but then of the droplets as "swollen aerosol particles". I think it's important to be clear throughout the paper what this layer is – is it haze (I.e. unactivated solution droplets) or is it fog (activated cloud droplets)?

- We apologise for the confusion, the layer below the sea ice cloud was taken to be a haze layer, with swollen aerosol particles/unactivated solution drops. This has been updated throughout the manuscript (Sect. 4.1.1, page 8, line 5; Sect. 4.2, page 10, line 30).

10. Section 4.3 and 2.2 – I think it is better to use a phrase like sea ice fraction rather than "ice concentration" (which makes reader think of snow particles in the clouds).

- "*Sea ice concentration*" has been changed to "*sea ice fraction*" throughout the manuscript to address the reviewer's concern (Sect. 2.2, page 5, lines 13, 15; Sect. 3, page 5, line 23; Sect. 4.3, page 11, line 13; Table 1; Figure 1).

11. Section 5.1 – talk about cirrus cloud and then "the possibility of another higher cloud layer" – makes it sound like there is a cloud higher than the cirrus, when in fact you are suggesting one in-between I think. Rephrase.

- The suggestion was that the ice crystals observed during run 5 could be precipitation from the cirrus cloud layers observed at high altitudes by the lidar and Modis. We are suggesting precipitation from the cirrus, not a higher cloud or one in between. This has been updated to the following in the manuscript: "*This suggests that precipitation from above was being sampled, which could possibly be from the high altitude cirrus layer observed closer to Spitsbergen*" (Sect. 5.1, page 12, lines 14-15).

Page 12, line 7: I would prefer dramatically to be replaced by something more quantitative.

- Addressed in manuscript, and subsequent sentence reworded to accommodate change in language. "*The most significant change in cloud microphysics over the transition is in the liquid phase, where the liquid water content and mean droplet size increase from 0.1 g/m$^3$ to 0.4 g/m$^3$ and 5 μm to 10 μm respectively.*" (Sect. 5.2, page 14, lines 7-9).

Page 14, line 17, is it appropriate to call this cloud cirrus if it has LWC? Maybe this comment is more directed at the Verlinde et al. (2007) paper, but I think it is also relevant for this paper.

- This reference was made in error. In fact, Verlinde et al. (2007) called the cloud above the one compared here a cirrus cloud: this cirrus cloud was glaciated as expected, yet there was liquid layer below which was being fed with ice by the cirrus layer. This change has been made clear in the manuscript; the overall message of the paragraph is unchanged, but the language has been rectified. The first paragraph of Section 5.4 (page 17, lines 10-14) has been rewritten to address these changes.

**Additional changes:**

The following changes were made to the manuscript out with the reviewers' comments. These were due to minor errors noticed on read-through.

1. "the ISDAC campaign" or "the M-PACE campaign" was often stated; however, this does not make sense due to what ISDAC and M-PACE stand for (Indirect and Semi-Direct Campaign and Mixed-Phase Arctic Cloud Experiment). Therefore, these references were changed to something more suitable in the manuscript.

2. Updates were made to improve readability with the requested content changes.

**Bibliography**

**DeMott**, P. J., Prenni, A. J., McMeeking, G. R., Sullivan, R. C., Petters, M. D., Tobo, Y., Niemand, M., Möhler, O., Snider, J. R., Wang, Z., and Kreidenweis, S. M.: Integrating laboratory and field data to quantify the immersion freezing ice nucleation activity of mineral dust particles, Atmospheric Chemistry & Physics, 15, 393–409, doi:10.5194/acp-15-393-2015, 2015.

**Griggs**, D. J. and **Choularton**, T. W.: A laboratory study of secondary ice particle production by the fragmentation of rime and vapour-grown ice crystals, Quarterly Journal of the Royal Meteorological Society, 112, 149–163, doi:10.1002/qj.49711247109, 1986.

**Jackson**, R. C., McFarquhar, G. M., Korolev, A. V., Earle, M. E., Liu, P. S. K., Lawson, R. P., Brooks, S., Wolde, M., Laskin, A., and Freer, M.: The dependence of ice microphysics on aerosol concentration in arctic mixed-phase stratus clouds during ISDAC and M-PACE, Journal of Geophysical Research (Atmospheres), 117, D15207, doi:10.1029/2012JD017668, 2012.

**McFarquhar**, G. M., Um, J., and Jackson, R.: Small Cloud Particle Shapes in Mixed-Phase Clouds, Journal of Applied Meteorology and Climatology, 52, 1277–1293, doi:10.1175/JAMC-D-12-0114.1, 2013.

**Miloshevich**, L. M., Paukkunen, A., Vömel, H., and Oltmans, S. J.: Development and Validation of a Time-Lag Correction for Vaisala Radiosonde Humidity Measurements, Journal of Atmospheric and Oceanic Technology, 21, 1305–1327, doi:10.1175/1520-0426(2004)021<1305:DAVOAT>2.0.CO;2, 2004.

**Poberaj**, G., Fix, A., Assion, A., Wirth, M., Kiemle, C., and Ehret, G.: Airborne all-solid-state DIAL for water vapour measurements in the tropopause region: system description and assessment of accuracy, Applied Physics B, 75, 165–172, doi:10.1007/s00340-002-0965-x, 2002.

**Ralph**, F. M., Neiman, P. J., and Rotunno, R.: Dropsonde Observations in Low-Level Jets over the Northeastern Pacific Ocean from CALJET-1998 and PACJET-2001: Mean Vertical-Profile and Atmospheric-River Characteristics, Monthly Weather Review, 133, 889–910, doi:10.1175/MWR2896.1, 2005.

**Tobo**, Y., Prenni, A. J., DeMott, P. J., Huffman, J. A., McCluskey, C. S., Tian, G., Pöhlker, C., Pöschl, U., and Kreidenweis, S. M.: Biological aerosol particles as a key determinant of ice nuclei populations in a forest ecosystem, Journal of Geophysical Research: Atmospheres, 118, 10 100–10 110, doi:10.1002/jgrd.50801, 2013.

**Verlinde**, J., Harrington, J. Y., McFarquhar, G. M., Yannuzzi, V. T., Avramov, A., Greenberg, S., Johnson, N., Zhang, G., Poellot, M. R., Mather, J. H., Turner, D. D., Eloranta, E. W., Zak, B. D., Prenni, A. J., Daniel, J. S., Kok, G. L., Tobin, D. C., Holz, R., Sassen, K., Spangenberg, D., Minnis, P., Tooman, T. P., Ivey, M. D., Richardson, S. J., Bahrmann, C. P., Shupe, M., Demott, P. J., Heymsfield, A. J., and Schofield, R.: The Mixed-Phase Arctic Cloud Experiment, Bulletin of the American Meteorological Society, 88, 205, doi:10.1175/BAMS-88-2-205, 2007.

**Wang**, J.: Evaluation of the Dropsonde Humidity Sensor Using Data from DYCOMS-II and IHOP_2002, Journal of Atmospheric and Oceanic Technology, 22, 247–257, doi:10.1175/JTECH1698.1, 2005.